# Water stress induced breakdown of carbon-water relations: indicators from diurnal FLUXNET patterns

Jacob A. Nelson, Nuno Carvalhais, Mirco Migliavacca, Markus Reichstein, Martin Jung

**Abstract**

Understanding of terrestrial carbon and water cycles is currently hampered by an uncertainty in how to capture the large variety of plant responses to drought. In FLUXNET, the global network of CO2 and H2O flux observations, many sites do not uniformly report the ancillary variables needed to study drought response physiology. To this end, we outline two data-driven indicators based on diurnal energy, water, and carbon flux patterns derived directly from the eddy covariance data and based on theorized physiological responses to hydraulic and non-stomatal limitations. Hydraulic limitations (i.e. intra-plant limitations to water movement) are proxied using the relative diurnal centroid ($C_{ET}^*$), which measures the degree to which the flux of evapotranspiration (ET) is shifted toward the morning. Non-stomatal limitations (e.g. inhibitions of biochemical reactions, RuBisCO activity, and/or mesophyll conductance) are characterized by the Diurnal Water:Carbon Index (DWCI), which measures the degree of coupling between ET and gross primary productivity (GPP) within each day. As a proof of concept we show the response of the metrics at 6 European sites during the 2003 heatwave event, showing varied response of morning shifts and decoupling. Globally, we found indications of hydraulic limitations in the form of significantly high frequencies of morning shifted days in dry/Mediterranean climates and savanna/evergreen plant functional types (PFT), whereas high frequencies of decoupling were dominated by dry climates and grassland/savanna PFTs indicating a prevalence of non-stomatal limitations in these ecosystems. Overall, both the diurnal centroid and DWCI were associated with high net radiation and low latent energy typical of drought. Using three water use efficiency (WUE) models, we found the mean differences between expected and observed WUE to be -0.09 to 0.44 umol/mmol and -0.29 to -0.40 umol/mmol for decoupled and morning shifted days respectively compared to mean differences -1.41 to -1.42 umol/mmol in dry conditions, suggesting that morning shifts/hydraulic responses are associated with an increase in WUE whereas decoupling/non-stomatal limitations are not.

## Introduction

Processes such as photosynthesis and transpiration are so intimately linked that knowledge and assumptions about one process are needed to accurately understand the other. Unfortunately, the relationship between carbon and water cycles

is not fully understood [47], passing the biases and uncertainties caused by an incomplete carbon:water framework back onto flux estimates specifically and global water and carbon cycle interactions and dynamics in general [18, 46, 15]. One source of uncertainty that is increasingly being identified is the diverse responses of plants to water limitation [55, 9, 44], which hampers the understanding and predictability of water and carbon cycles during drought. Here we outline potential causes of uncertainty in carbon:water dynamics in an effort to outline data-derived inductors based on current theory.

Classically, vegetation water and carbon fluxes are linked by stomates, where an open stomate allows CO2 to enter the leaf and, consequentially, water is lost. Most theoretical frameworks make some form of assumption that carbon assimilation (A) and water losses (T) are both contingent primarily on leaf stomatal conductance (gs). This assumed relationship allows us to pass between the realms of carbon and water, based on the assumption that at any given time both A and T are proportional to the stomatal conductance multiplied by the difference in internal and external CO2 and water vapor concentrations. More specifically,

$$A = g_s \cdot \Delta c \quad and \quad T = 1.6 \cdot g_s \cdot \Delta v \tag{1}$$

where $\Delta$c and $\Delta$v are the differences in inner and outer stomatal cavity concentrations of CO2 and water vapor, respectively. These diffusion equations lead to the relatively consistent carbon:water ratio, generally expressed as a water use efficiency ($WUE = A/T$). At the ecosystem level where direct measurements of A and T are not available, WUE is simply calculated as the ratio of gross primary productivity (GPP) to total evapotranspiration (ET) [22]. These carbon:water links are fundamental to understanding how stomata are regulated and underly key functioning in mechanistic plant and ecosystem models. One such set of models are those based on optimality theory which posit that plants tend to optimize carbon gains to water losses, such as those described by Katul et al. [17] and Katul, Palmroth, and Oren [16]. These concepts from Katul, which carry the assumptions of RuBisCO (light) limitation, were built upon by Zhou et al. [54] and Zhou et al. [52] to give the equation,

$$uWUE = \frac{GPP \cdot \sqrt{VPD}}{ET} \tag{2}$$

where the $\sqrt{VPD}$ accounts for the stomatal response to vapor pressure deficit (VPD) assuming stomatal response optimizes carbon gain to water losses. Accounting for the VPD response allows for a more stable metric of WUE that is temporally more stable and physiologically more meaningful, such as when comparing the diurnal cycles of carbon and water. As ET is the sum of both T and non-biological evaporation (e.g. soil and intercepted evaporation), often periods during and shortly after rain events are excluded from WUE estimates to minimize the influence of non-plant evapora-

tion. Ultimately, calculations of WUE provide a simple summary of the cost in water per carbon gain and becomes an indicator for how plants have and will adapt to the physical limitations of their changing environments [18, 47].

Though assuming a rigid carbon:water relationship works well in conditions when ecosystems are moderately wet, conditions associated with the majority of carbon and water fluxes, an inflexible carbon:water assumption is unsatisfactory in that these assumptions may breakdown as plants shift from light to water limitations. Indeed, in a review of leaf level stomatal conductance models, Damour et al. [7] concluded that the majority of stomatal models fail to adequately capture the effects of drought. This failure to capture the effects of drought is not only disconcerting as water limited conditions are when ecosystems are most at risk, but an incomplete framework tends to propagates errors and uncertainties from models into estimates of the water and carbon cycles. For instance, in outlining a road map for improved modeling of photosynthesis, Rogers et al. [44] noted as key recommendations both improving information about water:carbon relations (in the form of the stomatal slope parameter $g_1$) as well as improving understanding of the response of carbon assimilation to drought. Similarly, in an analysis of parameter uncertainties for a terrestrial biosphere model, Dietze et al. [9] found that two of the top five parameters contributing to the predictive uncertainty of net primary productivity were associated with plant water regulation. This uncertainty is reflected in the stomatal conductance parameterization exercise from Knauer, Werner, and Zaehle [19], where the authors were able to improve model performance in predicting EC measured GPP and ET by including atmospheric effects (in the form of VPD) on stomatal conductance, but concluded that further improvement required global understanding of water limitation response variation across plant functional traits and growing conditions, which is currently unavailable.

Two ideas to account for the errors in carbon:water assumptions under dry conditions have begun to emerge: that hydraulic limitations in transporting water from root to leaf change stomatal responses and thus limit transpiration under high demand, or that changes in the intra-leaf processes of carbon transport and fixation under drought conditions result in non-stomatal limitations that impact carbon assimilation independently of water fluxes [34].

As soil water potentials in the root zone become increasingly negative, the long-term plant strategy may turn from optimizing carbon fixation to preventing damage to hydraulic architecture [48]. As such, stomata and transpiration are likely to increasingly respond not just to atmospheric conditions, but also soil moisture. Under this hydraulic limitation framework, a plant will be reacting to the inability to transport water, even though the key control mechanism for a plant is via the stomata, possibly expressed as an increase in sensitivity. Such assumptions are consistent with the mechanisms encoded in some land surface and ecosystem models, which account for water limitations by scaling the water to carbon ratio in relation to available soil moisture. Though this method should link the leaf physiology to the soil and thus capture some hydraulic limitation, it has been criticized for not capturing the variety of drought responses found in different plant species and ecosystems [8]. This diversity in plant responses has been pointed to as a key point of uncertainty in earth system models [9].

Though ecosystem water and carbon fluxes are predominantly controlled by stomates, non-stomatal or bio/photo-chemical inhibitions to carbon assimilation are worth considering as they have the capacity to decouple the water-carbon exchange. This decoupling could include conditions where the stomates are transpiring water but intra-leaf factors are slowing carbon fixation, changing the intrinsic water use efficiency directly. Intra-leaf factors could include effects such as production of reactive oxygen species [23]; environmental limitations to the photosynthetic pathways, such as leaf temperature [31]; or declines in mesophyll conductance [11]. Non-stomatal limitations have been observed at ecosystem scale [42, 32], though the exact mechanism is difficult to elucidate [40]. These effects likely vary between species, as well as with the rate of onset of drought, access to water, and other environmental conditions.

**Objectives**

There seems to be a collective conclusion that the breakdown of carbon:water assumptions needs to be better character-ized in general, and specifically for implementation in modeling frameworks [De Kauwe et al. [8]; Manzoni [25]; Zhou et al. [55]; Flexas et al. [11]; Egea, Verhoef, and Vidale [10];]. Though the problem is becoming clear, the way forward is hampered by an uncertainty in how to capture the large variety in the response to drought across climates, strate-gies, and species. In this sense, the use of EC measured diurnal patterns of carbon, water, and energy fluxes to derive clues on ecosystem drought responses at a daily resolution could prove valuable both as a means to identify potential periods of ecosystem stress, inform machine learning algorithms on ecophysiological conditions not found in environ-mental variables, as well as benchmarking a models ability to capture sub-daily dynamics. To this end, we propose two data-driven indicators of water stress, the diurnal water:carbon index (DWCI) and the relative diurnal centroid in LE ($C_{ET}^*$). Both metrics are derived directly from the EC data and based on expected physiological responses to hydraulic and non-stomatal limitations. Using these data-driven indicators we then characterize the distribution of these limita-tions across a global spread of climate and vegetation types. Finally, we explore the ability of these indicators to detect the disagreements between modeled and observed water use efficiency, and explore how these biases may be attributed to hydraulic and non-stomatal limitations.

## Methods and Materials

**Data**

Carbon, water, and every fluxes measured with EC, as well as meteorological data, were obtained from the 2007 FLUXNET La Thuile Synthesis Dataset [12]. Half-hourly latent heat and net ecosystem exchange (NEE) fluxes were collected and processed using standard QA/QC procedures [36], gap-filling and partitioning algorithms [41]. From the

database, half-hourly gross primary productivity (GPP) and ET data (derived from latent heat flux measurements) were downloaded and used for the following analysis. An interactive map of sites used can be found in File S1.

In order to provide a consistent measure of ecosystem dryness that can be utilized across sites, the ratio of water evaporated to potential water evaporated was calculated as evaporative fraction (EF), or the fraction of actual ET to Potential ET (PET). PET was calculated as the daily fraction between the measured ET and estimated ET via a Priestly-Taylor model [38] using site measured net radiation (Rn) and air temperature ($T_{air}$). The slope (alpha parameter) was fit for each site-year using 95th quantile regression [20] instead of using the original 1.26 value derived for a "well watered crop" [38].

In order to get high quality data and minimize the influence of abiotic evaporation (hereafter just evaporation), all data was filtered with the aim to include only non-gap filled data in the growing season with dry surface conditions. Growing season was defined as all days where GPP > 1 $gC \cdot m^{-2} \cdot d^{-1}$ and daily mean air temperature > 5 °C. These threshold were shown to give good response in the proposed metrics while minimizing variability due to low diurnal signals, a sensitivity analysis of which can be found in supplementary Figure S2. In an effort to minimize contributions of evaporation, the conservative soil wetness index (CSWI) was employed which was designed to estimate whether the ecosystem is likely to have "dry" surfaces and therefore ET is likely to be dominated by transpiration. This approach requires a certain amount of evaporation to occur after a rain event before the surface is considered to be "dry" and can be contrasted to the method of removing a set time period after rain employed in previous studies [30, 2, 18]. CSWI is calculating by first quantifying the storage at time t ($S_t$) as,

$$S_t = min\left(S_{t-1} + P_t - ET_t, S_o\right) \tag{3}$$

where $ET_t$ and $P_t$ are the ET and precipitation at time-step t respectively, $S_t$ is effectively capped at a maximum storage value of $S_o$, which was set to 5 mm. Furthermore, to make the metric conservative in regards to assumed water inputs, any precipitation event will refill the storage from 0 mm,

$$CSWI = max\left(S_t, min\left(P_t, S_o\right)\right) \tag{4}$$

which has the effect of requiring all precipitation up to 5 mm to be evaporated from the system before negative storage can occur. Any gaps in the precipitation data were assumed to be a precipitation event of 5 mm in order to prevent any unmeasured precipitation from biasing the results by inadvertently including rainy days. Code and further outline of the algorithm can be found in File S3 as well as at Nelson [33]. Evaporation was assumed to be negligible when $CSWI<0$. This method was used over the more standard method of removing 1-5 days after a rain event, as it does not

make the assumption that the surface will dry in a fixed amount of time, instead relying on a minimum amount of ET.

As a comparison, the median time period for the CSWI to go from fully wet (CSWI=5) to "dry" (CSWI<=0) was 3.5 days

across all sites in summer, where summer was defined as the period when daily potential radiation above median daily

potential radiation for each site.

The data filtering as outlined in this section was designed to isolate periods firmly in the growing season when plants are

active and the signal of ET is most likely to be dominated by plant controls.

## Relative diurnal centroid ($C_{ET}^*$)

As soils dry, it becomes more difficult to transport stem and root zone moisture to the leaf, potentially causing hydraulic

limitations for the plant to transport water. This shift was seen in eddy covariance data in a study by Wilson et al. [50],

who examined the shift of latent compared to sensible heat, which suggested that a shift in water fluxes towards dawn

can be indicative of afternoon stomatal closure. Shifts were further explored in a modeling study by Matheny et al. [29]

which found that the morning shift was not well captured by models and attributed the errors to inadequate hydraulic

limitations in the models. The daily cycle of wetting and drying acts as a capacitor in the hydraulic circuit, allowing water

stores to be more easily transported in the morning and depleting in the afternoon. As bulk soil moisture declines, this

effect may be strong enough to shift the diurnal cycle of ET significantly toward the morning. Quantifying diurnal shifts

in EC data using the diurnal centroid was first explored by Wilson et al. [50]: defined as the flux weighted mean hour, or

$$C_{flux} = \frac{\sum flux_t \cdot t}{\sum flux_t} \tag{5}$$

where t is a regular, sub-daily time interval (here t measures as decimal hour at half-hourly time-step). The resulting

$C_{flux}$ is the weighted mean hour of the diurnal cycle of that particular flux for that particular day. For example, if a

calculated $C_{ET}$ for a given day (using measurements of decimal hour) equals 12.25, this would entail that the weighted

mean for that day is 15 minutes past noon. Figure 1 shows an example of the shifts in the monthly average cycle from

a wet month to a dry month. In order to isolate a shift, we then had to control for variations in global radiation (Rg),

both fluctuations due to clouds and differences in the timing of solar noon. Therefore, the difference between the diurnal

centroids of ET ($C_{ET}$) and Rg ($C_{Rg}$) was calculated as

$$C_{ET}^* = C_{Rg} - C_{ET} \tag{6}$$

giving $C_{ET}^*$ as the diurnal centroid of ET relative to Rg. The resulting values of $C_{ET}^*$ are not tied to the carbon cycle, which

<sup>163</sup> can be affected by non-stomatal limitations and generally shows a more prominent midday depression. Annotated code

<sup>164</sup> for the CSWI calculation can be found in File S4 as well as at Nelson [33]. Though a diurnal centroid can be calculated

<sup>165</sup> for any diurnal cycle, basing a metric on the morning shift of ET relative to Rg has the advantage of targeting the non-

<sup>166</sup> atmospheric drivers of the water flux, of which there are few ancillary variables.

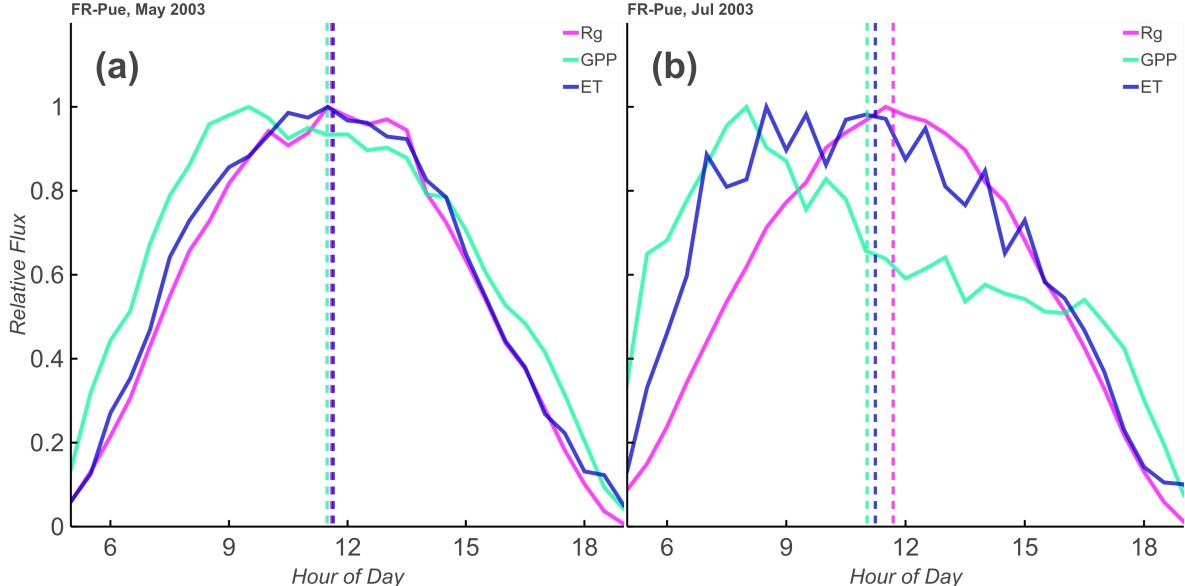

Figure 1: One month average cycle (soild lines) and accompanying diurnal centroid (vertical dashed lines) of incoming shortwave radiation (Rg), evapotranspiration (ET), and gross primary productivity (GPP) at the Peuchabon, France ('FR-Pue') site during 2003. May is relatively wet (32 mm rainfall, left) and July is relatively dry (0 mm rainfall, right). While ET and Rg correspond well in the wet month, the dry month shows a distinct phase shift in both GPP and ET fluxes towards the morning, as well as a midday depression in GPP.

<sup>167</sup> **Diurnal water carbon index (DWCI)**

<sup>168</sup> If transpiration and carbon assimilation are predominantly controlled by stomatal conductance, it follows that their

<sup>169</sup> diurnal cycles should be largely in sync. In other words, regardless of a plants maximum T or A, if the stomates start to

<sup>170</sup> close, both rates should be decrease by a similar percentage. On the other hand, non-stomatal limitations that inhibit

<sup>171</sup> carbon assimilation independent of water have the capability to alter the diurnal cycle on just one flux, causing them to

<sup>172</sup> decouple. In an effort to quantify the degree of carbon:water coupling for an individual day, we examined the relationship

<sup>173</sup> of GPP and ET, where,

$$ET \propto GPP \cdot \sqrt{VPD} \tag{7}$$

<sup>174</sup> or,

$$ET = i \cdot GPP \cdot \sqrt{VPD} \tag{8}$$

This relationship incorporates the assumption that, at least over short time scales, the amount of carbon that enters the leaf is proportional to the amount of water that leaves, and also incorporates the non-linear response of stomates to VPD [17, 16, 54]. This model, though simple, has been shown to work well across a variety of EC sites [52]. Figure 2 (upper panels a,b) shows a comparison between the daily cycles in a wet and dry month. By calculating a daily correlation between the normalized daily cycles of ET and $GPP \cdot \sqrt{VPD}$, we come to a correlation coefficient for each day (see Figure 2, lower panels c,d). For well watered days in the growing season the two signals tend to be well correlated ($\rho > 0.9$), but tends to be less correlated in periods of stress, a comparison of which can be seen seen in Figure 2 (lower).

As it is, this daily correlation coefficient is dependent on the signal strength, or magnitude, of the flux. Low correlation values could just as easily be from carbon:water decoupling as to a low signal to noise ratio. Therefore, to produce a more robust metric and account for these statistical decreases in correlation, we turned the daily correlation coefficient into an index based on its rank in a distribution of correlation coefficients from artificial datasets. These artificial datasets are constructed using the diurnal signal from potential radiation, with Gaussian noise ($\mathcal{N}(0, \sigma)$) added according to the standard deviation random uncertainty of the ET and NEE fluxes, or

$$LE_{artificial} = \frac{Rg_{pot}}{\overline{Rg_{pot}}} \cdot \overline{LE} + \mathcal{N}(0, \sigma^2_{LE|NEE}) \tag{9}$$

and

$$NEE_{artificial} = \frac{Rg_{pot}}{\overline{Rg_{pot}}} \cdot \overline{NEE} + \mathcal{N}(0, \sigma^2_{NEE|LE}) \tag{10}$$

Uncertainties of the NEE and ET fluxes were estimated from the gap filling procedure of Reichstein et al. [41], with the uncertainty equal to the standard deviation of flux measurements within a time window and similar meteorological conditions. As GPP is calculated from gap-filled values of NEE, the uncertainty from NEE was used for GPP. Furthermore, the correlation structure between the noises in LE and and NEE was preserved in the artificial dataset.

In essence, by using the underlying signal from potential radiation, both the artificial ET and $GPP \cdot \sqrt{VPD}$ are perfectly correlated when no noise is added. Adding noise then isolates the decoupling effect of signal to noise ratio. An artificial correlation coefficient can then be calculated from the two artificial datasets in the same manner as from the real dataset, and this experiment is repeated 100 times for each day, giving a daily distribution of artificial correlation coefficients. The rank of the real correlation coefficient in the distribution from the artificial set gives a probability that the carbon and

water signals are actually coupled. The resulting index has a range of 0-100, with 100 indicating that the real correlation coefficient was greater than the entire artificial set, and therefore it is very likely that carbon and water are coupled. From this index we can now quantify if the water and carbon signals are coupled for any given day, and therefore shed light onto whether the two fluxes are only controlled by the opening and closing of stomates. Annotated code for this calculation can be found in File S5 as well as at Nelson [33].

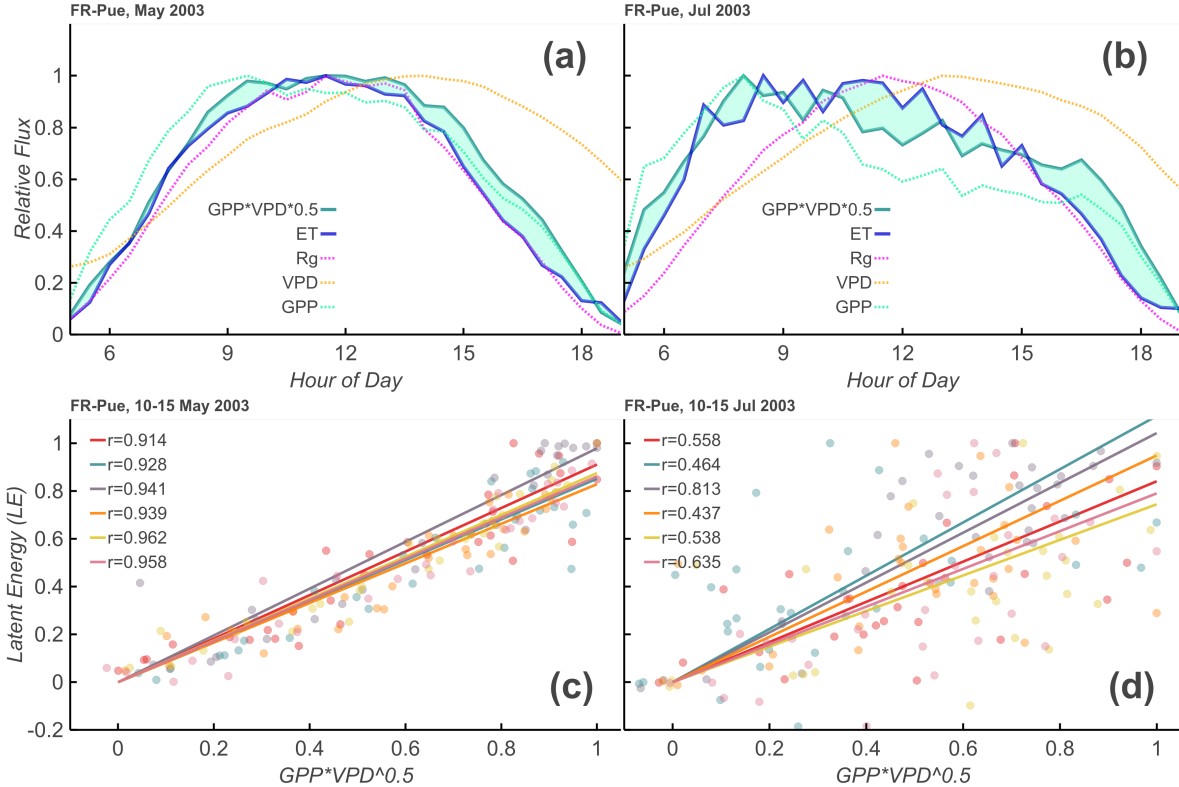

Figure 2: Theoretical overview of diurnal water carbon index upper panels: One month average diurnal cycle of incoming shortwave radiation (Rg), evapotranspiration (ET), vapor pressure deficit (VPD), gross primary productivity (GPP), and $GPP*VPD^{-0.5}$ at the Peuchabon Forest, France ('FR-Pue') site during 2003. Discrepencies between $GPP*VPD^{-0.5}$ and ET increase from the relatively wet May (32 mm rainfall, left) to the relatively dry July (0 mm rainfall, right). lower panels: These discrepencies are reflected in the daily correlation values between $GPP*VPD^{-0.5}$ and ET, giving an indication of the appropriatness of the uWUE model for each day, as well as the degree of coupling between water and carbon signals.

**Models and parameter estimation**

In order to benchmark whether these metrics are capturing information that is possibly not being captured in modern model frameworks, three simple models were used to estimate WUE (GPP/ET) for each day at each site and compared to actual flux data. The purpose of the exercise was to evaluate if bias in the model predictions were associated with decoupled or morning shifted days, thus indicating that the metrics are corresponding to information that the models are unable to capture. Here we utilize three models to provide a spectrum of theoretical to empirical basis. The "Katul-

Zhou" model, as defined and used in calculation of the DWCI, is based in stomatal optimization theory [17, 16, 52], which makes the assumption that the WUE is constant if corrected by the effect of VPD, using an inverse square root as the assumed relationship. Though the constant nature of uWUE may not be correct, with the optimal carbon cost of water changing over day or weeks [26, 35], a yearly parameter of uWUE was estimated which is consistent with other modeling exercises [53]. One step away from a theoretical basis is a revision of this model by Boese et al. [3], the "Boese" model, where an additional radiation term was added such that,

$$ET = i \cdot GPP \cdot \sqrt{VPD} + r \cdot Rg \tag{11}$$

where i and r are parameters fit to each site-year. This relationship with Rg was shown to have a better predictive performance for EC data from 115 sites [3]. The interpretation of this extra radiation term is not clear and is difficult to reconcile with the current understanding of physiology. It is possible the term could be related to biophysical effects, e.g. VPD at leaf surface vs the measured ambient VPD. Nevertheless, the Boese model is an empirical and ecosystem scale model that complements the theoretical and originally leaf-level model from Katul-Zhou.

Parameters of these models were estimated for each site-year. The Boese model parameters were fit using trimmed least squares regression (TLS) which minimizes the 90th percentile of SSE to prevent influence of large outliers [45, 43]. As the error in both ET and GPP are assumed to be of similar magnitude, the i parameter in the Katul-Zhou model was calculated using geometric mean regression, where the final slope was calculated as the geometric mean of the parameters from

$$ET = i_{GPP} \cdot GPP \cdot \sqrt{VPD} \ and \ GPP \cdot \sqrt{VPD} = \frac{ET}{i_{ET}} \tag{12}$$

Both the Katul-Zhou and Boese models are theoretically based and here implemented have the underlying assumptions of RuBisCO-limited conditions and constant carbon cost of water throughout the season which may not reflect reality. Therefore a fully empirical and highly non-linear model can give insight into how much information is actually stored in the data while minimizing any assumptions. As a fully empirical model, a random forest regression (RandomForestRegressor from Pedregosa et al. [37] based on Breiman [4]) was fit to half-hourly ET data for each site using Rg, VPD, $Tair$, GPP and year as input parameters. Values were estimated using 50 trees with predictions made using out-of-bag estimates to prevent over-fitted model predictions.

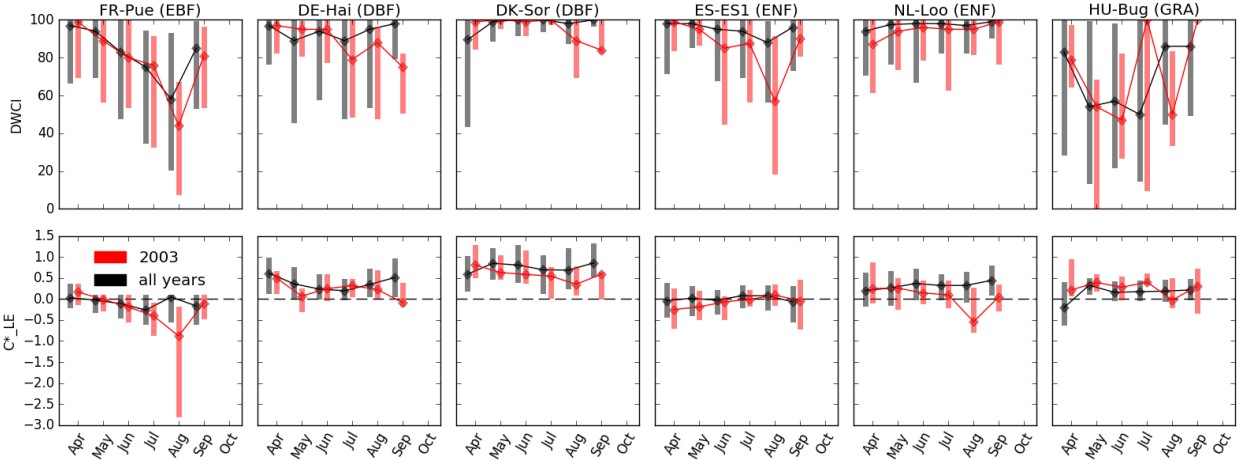

Figure 3: Monthly median diurnal water carbon index (DWCI, lower panels) and diurnal centroids ($C_{ET}^*$, upper panels) for 6 sites in Europe. Data from all years available (black) is compared to 2003 (red) during which a drought event resulted in high temperatures and low precipitation throughout the summer. Note DWCI of 0-100 indicate lowest-highest probability of diurnal carbon:water coupling and $C_{ET}^*$ of -1 to 1 indicate one hour morning shifted to one hour afternoon shifted ET. Vertical bars represent interquartile range. Sites from 4 plant functional types: evergreen broadleaf (EBF), deciduous broadleaf (DBF) and evergeen needleleaf (ENF) forests, as well as grasslands (GRA). Ecosystems show tendancies of morning shifts (e.g. DK-Sor and NL-Loo) and carbon:water decoupling (e.g. ES-ES1 and HU-Bug) during the drought year.

## Results

As a case study, $C_{ET}^*$ and DWCI time-courses for six sites from Europe are shown in Figure 3, with an emphasis on 2003 when the continent was struck by a heatwave that was shown to effect both the carbon and water cycles [6, 39, 13]. For DWCI, forest sties showed high water:carbon coupling throughout the growing season, with the exception of Peuchebon (FR-Pue) which showed a regular seasonal cycle of decoupling. The grassland site (HU-Bg) showed a higher variability in DWCI compared to the forest sites (all others). All sites showed either a decrease in median DWCI or an increase in variability during 2003, generally in July or August, particularly at Hainich (DE-Hai), Bugacpuszta (HU-Bug), and El Saler (ES-ES1). This increase in decoupling during 2003 is consistent with the hypothesis of non-stomatal limitations being expressed in hot, dry conditions which can affect carbon fixing mechanisms. Median diurnal centroid values across all years varied in absolute magnitude, but were generally near or above zero, i.e. the water cycle showed no shift or an afternoon shift. One exception would be the Mediterranean oak forest of Puechabon, which shows a slight seasonal cycle of morning shifts going from a slight afternoon shift to a slight morning shift during June, July, and August. During drought years, sites that showed distinctive morning shifts were Puechabon (FR-Pue), Soroe (DK-Sor), and Loobos (NL-Loo). The framework that morning shifts are associated with water stress from soil moisture depletion would be supported by the increase in morning shifts during 2003, though factors such as species composition and access to soil water would play a significant role and could account for the differences among sites. All sites had significantly different (p<0.05, Wilcoxon rank-sum test) DWCI values between 2003 and all other years except Puechabon, whereas

$_{248}$ with $C_{ET}^*$ only Puechabon, Soroe, and Loobos showed significant differences.

**$_{249}$ Distribution of data driven indicators by vegetation type and climate**

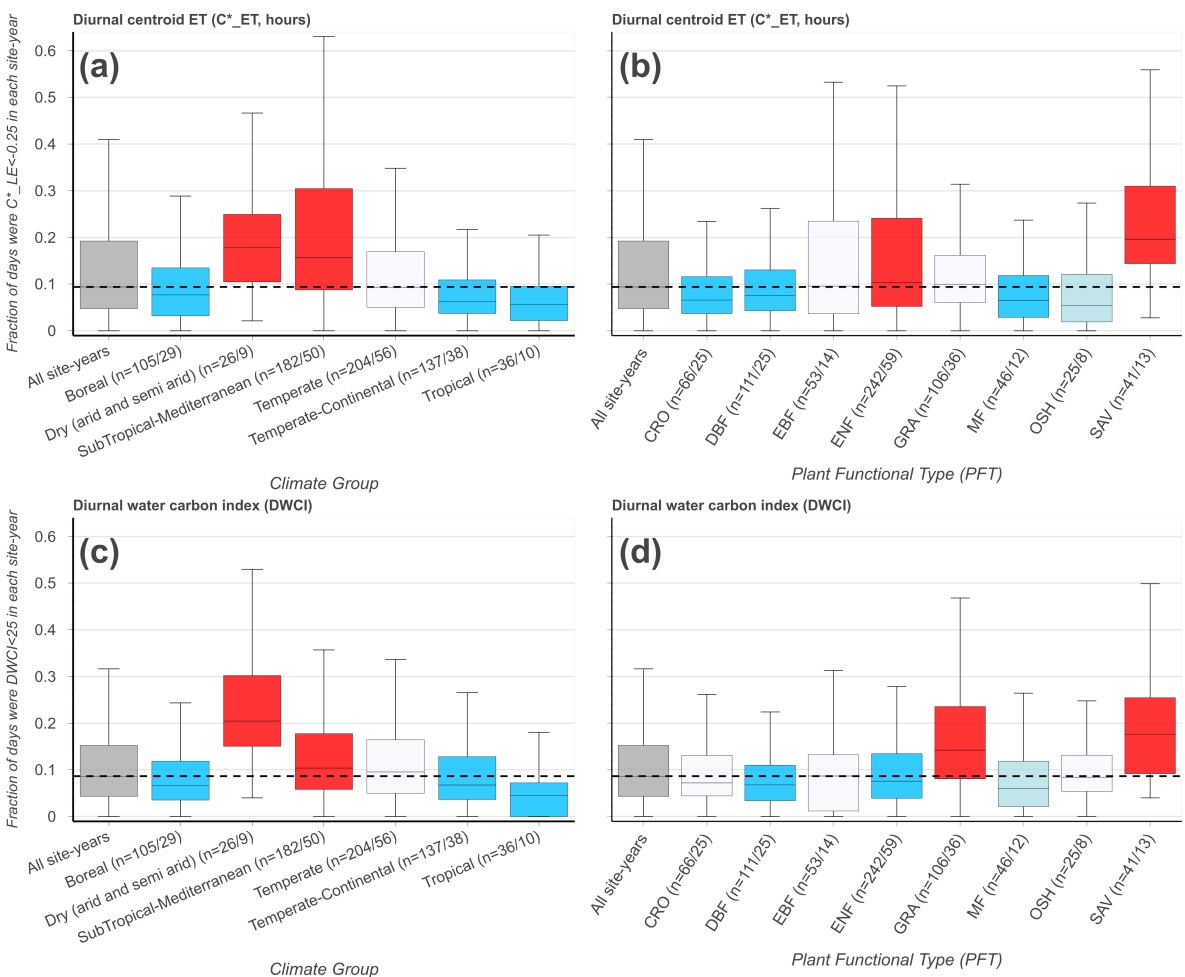

Figure 4: The frequency of morning-shifted Diurnal Centroids ($C_{ET}^*$<-0.25 hours, upper panels a,b) and low diurnal water carbon index (DWCI<25, lower panels c,d) for 690 fluxnet site-years/192 sites, grouped by climate group (left panels a,c) and plant functional type (right panels b,d). Group labels on x-axis indicate the number of site-years/sites (n=site-years/sites) for each category. Dashed line is the median for all site-years. Color shade indicates level of significance, with light colors and dark colors having p-values <0.10 and <0.05 respectively (Wilcoxon–Mann–Whitney two-sample rank-sum test), red and blue colors indicate distributions higher and lower respectively compared to data from all sites excluding the group. Only sites-years with at least 20 data points and groups with more than 5 site-years were included.

$_{250}$ The frequency of low values of diurnal centroid and DWCI across climate groups and plant functional types is shown

$_{251}$ in Figure 4. The thresholds designating decoupling and morning shifts were 25 and -0.25 for DWCI and $C_{ET}^*$ respec-

$_{252}$ tively. These thresholds were chosen to highlight frequency differences between sites and were shown to have large

$_{253}$ metric responses under dry conditions while having low frequencies under wetter conditions (see sensitivity analysis in

$_{254}$ supplementary Figure S2). Furthermore, these thresholds results in a similar median frequency of uncoupled and morn-

ing shifted days between all site-years being 8.7% and 9.4% of days respectively. The similarity in median frequencies across site-years allowed for easier inter-comparison between the two metrics. The frequency of decoupling and morning shifts using these thresholds for each site can be found in the map found in File S1. Though there is a fairly large variance across climate groups and plant functional types, low values of both DWCI and $C_{ET}^*$ occur at higher frequencies in savanna ecosystems and dry or Mediterranean climates. Conversely, lower frequencies of both metrics are seen in tropical, boreal, and temperate-continental climates. Strikingly, the arid and semi-arid climate group seems to be associated with the majority of low DWCI occurrences, with a median frequency of about 20% of days being uncoupled between site-years. Overall, frequencies were highly variable within plant functional types. Interestingly, $C_{ET}^*$ seems to be more variable in moderately dry ecosystems with potentially deep roots, favoring woodier savannas and evergreen needle-leaf forests over grasslands and open shrub lands. In contrast, DWCI shows similarly high frequencies from savannas and grasslands. The differing responses between tree and grass dominated ecosystems can be further seen in Figure 5, where savanna and grassland ecosystems show a distinct decrease in DWCI under conditions of low EF, in contrast to the forested sites which show a higher degree of carbon:water coupling, though still a slight decrease. Forested ecosystems show a higher degree of morning shift under low EF conditions when compared to grasslands, with savannas being somewhere between the two.

The response of both variables to drought stress is further observed in Figure 6, where low mean values of both DWCI and $C_{ET}^*$ are associated with conditions of high net radiation and low latent energy, indicative of drought. As this figure includes all days from all sites which meet the filtering outlined in the Data subsection of the Methods, i.e. dry periods in the growing season, these figures exhibit the universality of the metrics across climates, ecosystems, and time periods. This pattern is much cleaner with the diurnal centroid than with DWCI, though mean values are generally above 50 for most bins, indicating that most days are well coupled. Low values of both indicators are also seen under conditions with low Rn and high latent energy (as seen by the dark streak at the top edged in Figures 6c,e), which is generally not associated with drought stress. Further analysis showed that these points are also associated with energy balance over closure, where the sum of latent and sensible heat is greater than net radiation (ET+H>Rn, see Figure S2) and therefore likely represent a data problem rather than a physiological response. Removing all days where the energy balance is over closed did not alter the patterns associated with drought. Apart from the response to periods of high LE and low Rn, the metrics showed diverging response when looking at EF (ET/PET which is similar to LE/Rn) and VPD, with DWCI showing a much stronger response to VPD and $C_{ET}^*$ showing a much stronger response to EF (Figure 6a,d). This difference in response would indicate that DWCI is more responsive to atmospheric demand (estimated via VPD) and $C_{ET}^*$ is more responsive to water limitations. Both DWCI and $C_{ET}^*$ also show a trend with low GPP, although in the case of the diurnal centroid the effect is limited to both low GPP and ET (Figure 6c,g).

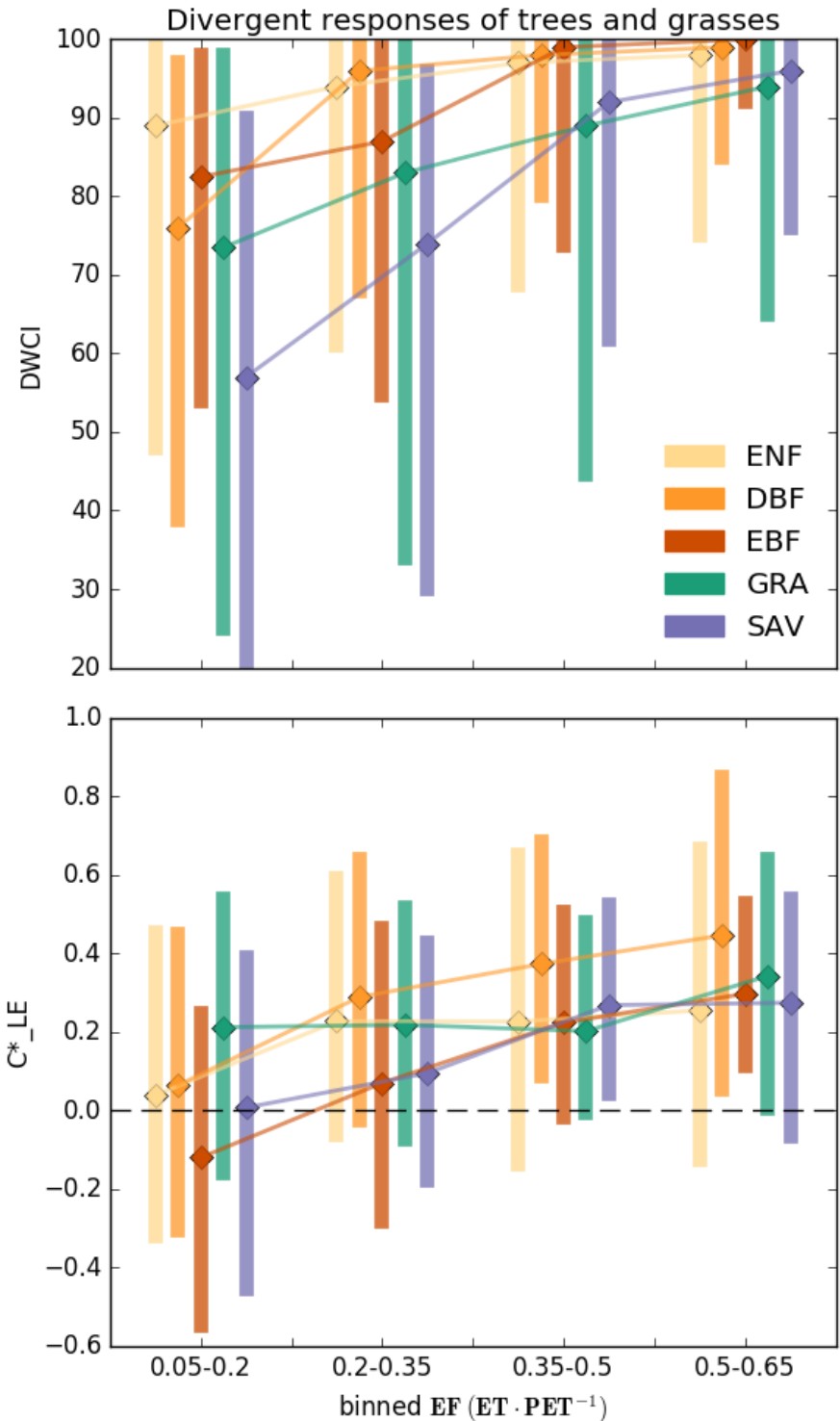

Figure 5: Median diurnal water carbon index (DWCI, upper panel) and diurnal centroid ($C^*_{ET}$, lower panel) of plant flunctional types binned by evaporative Fraction (EF, low values indicate dry conditions). Note DWCI of 0-100 indicate lowest-highest probability of diurnal carbon:water coupling and $C^*_{ET}$ of -1 to 1 indicate one hour morning shifted to one hour afternoon shifted ET. Evergreen needleleaf (ENF), deciduous broadleaf (DBF), and evergreen boradleaf (EBF) forests show increased morning shifts (low $C^*_{ET}$) with decreasing EF when compared to grassland (GRA) sites which tended to have decreased carbon:water decoupling (low DWCI) with decreasing EF. Savanna ecosystems (SAV) show a high degree of decoupling and intermediate levels of morning shifts. Vertical bars represent interquartile range.

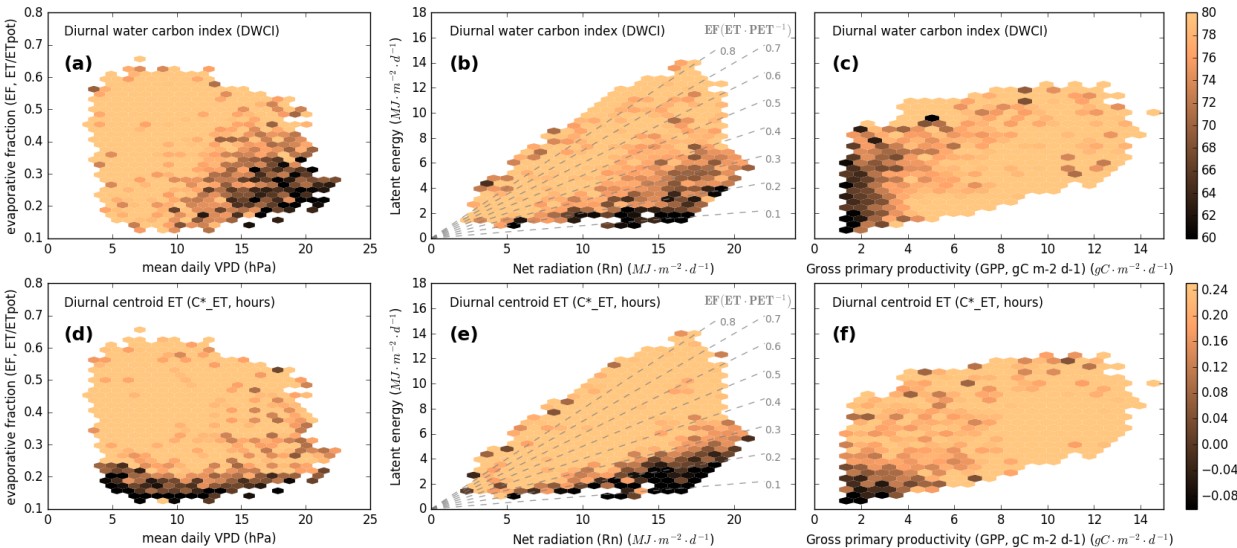

Figure 6: Mean DWCI (upper panels) and $C^*_{ET}$ (lower panels) with respect to evaporative fraction (EF) by vapor pressure deficit VPD (a,d), latent energy (LE) by Rn (b,e) and LE by GPP (c,g). Note DWCI of 0-100 indicate lowest-highest probability of diurnal carbon:water coupling and $C^*_{ET}$ of -1 to 1 indicate one hour morning shifted to one hour afternoon shifted ET. Points with high Rn and low LE are associated with both low DWCI and $C^*_{ET}$, indicating that both metrics are related to water limitations. Though both metrics are associated with low EF, DWCI shows a much higher response to atmospheric demand as measured by VPD, with $C^*_{ET}$ showing very limited response. Both metrics, and DWCI in particular, show low values with high ET and low Rn, though these points are also associated with over closed energy balances (LE+H>Rn). Both metrics are associated with low GPP, but the $C^*_{ET}$ is restricted to both low GPP and ET, indicating water and carbon can decouple over a wider range of water stress. This also holds when points with energy balance over-closer are excluded (data not shown).

**Difference between modeled and actual WUE**

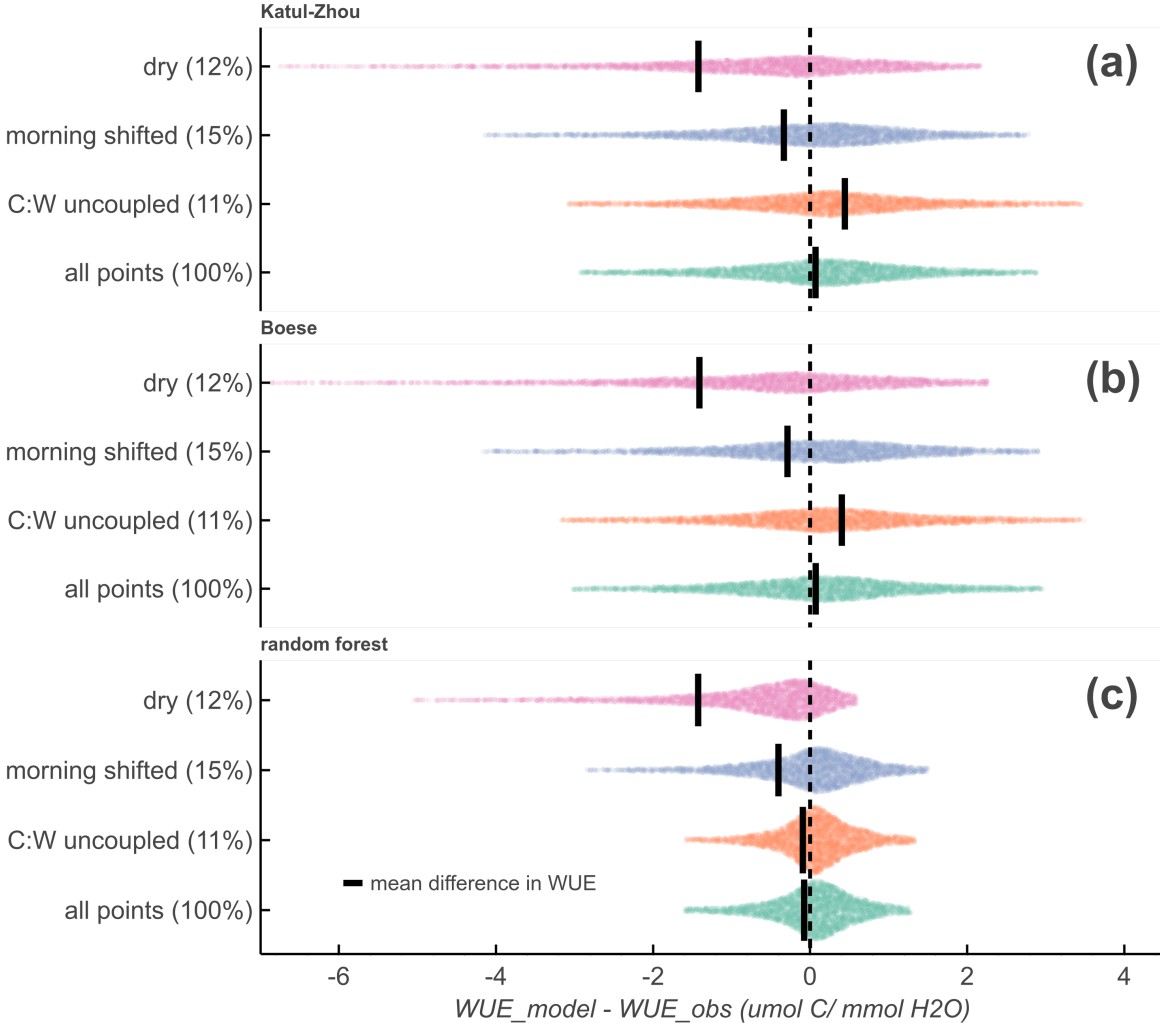

Figure 7: Difference in modeled and measured WUE for Katul-Zhou (a), Boese (b), and random forest (c) models. The random forest model was fit using Rg, VPD, $T_{air}$, GPP, and year. Thresholds designating dry, morning shifted, and C:W uncoupled days were EF<0.2, $C^*_{ET}$<-0.25, and DWCI<25 respectively for each day. The distributions span from the 5th to 95th percentiles, and the width of each gives an indication of the variance, which is larger in the sub groups compared to all points. Furthermore, the mean difference in WUE (black lines) tends to be shifted in dry and morning shifted days indicating a mean underestimation of WUE by the models mostly due to the long tails. Decoupled days show highter variance, but no clear pattern in under- or over-estimation. The percentage of days in each category are designated next to y-axis label in parenthesis.

Figure 7 shows the difference between expected and observed WUE from the Katul-Zhou, Boese, and random forest (RF) models, with respect to conditions of drought as characterized by low evaporative fraction (EF<0.2), C:W decoupling (DWCI<25), and morning shifts ($C^*_{ET}$<-0.25). This exercise was designed to test whether the metrics were associated with bias in the models, indicating that the metrics are able to capture information that the models are not (as further outlined in Methods and Materials subsection "models and parameter estimation"). For all models, the dry days show the largest average shift between expected and observed WUE, followed by morning shifted days. Uncoupled days show

the smallest shifts for all models, with an overestimation of WUE for the Katul-Zhou and Boese models and no significant shift of WUE with the random forest model. As all models were calibrated within a site-year, the over or under estimation of WUE indicate an inability of the model to capture a change in the system. Cases of mean mis-estimation tended to be influenced by long tails in the distribution with median differences being less exaggerated. However, these long tails are indicative of major model error in periods where the ecosystem is likely under stress conditions.

## Discussion

### Looking beyond sums and means

The proposed metrics, DWCI and $C_{ET}^*$, depart from more traditional methods to summarize from sub-daily to daily timescales such as sums and means. This departure is advantageous in that it extracts added information that may have been otherwise ignored by turning the focus from signal amplitude to the signal shape. However, these new metrics also come with their own set of caveats, most notably issues with data quality confounding interpretability. Both metrics are susceptible to noise, as one or two errant points within a day can be reflected as a decrease in correlation or a shift in diurnal centroid. This is evident from the existence of very afternoon shifted $C_{ET}^*$, sometimes by more than an hour, which the authors have no proposed explanation for other than noise in the data. However, attributing highly afternoon shifted points as poor data requires further investigation. Note here that the "resting" $C_{ET}^*$ seems to be slightly afternoon shifted, which could be caused by real physiological factors such as differences in the incoming SW radiation (Rg) used in the calculation and net radiation (Rn), higher atmospherics demand (VPD) in the afternoon driving higher ET, or increased convection throughout the day resulting in higher transport of water away from the canopy, and is likely a combination of all three. Differences in resting $C_{ET}^*$ between sites could also be from instrumental causes such as radiometric sensors which are not adequately leveled or dirty, though the consistent, slight afternoon shifts would suggest this is a real response. Despite the possible shortcomings, both metrics show a definite response to drought conditions across the broad array of sites, climates, and ecosystems contained in FLUXNET (see Figure 6), and give valuable insight into the underlying physiology. Given the broad nature of the analysis here, the metrics and hypothesis presented would benefit from site specific validations such as looking to see if the morning shits and decoupling are indeed associated with lower soil moisture levels, leaf water potentials, and/or decreases in sap flux. Sap flux in particular could give some interesting insights, as the diurnal patters in sap flux velocity will also have an offset to incoming radiation related to tree capacitance, therefore relating sap flow diurnal centroids to the ET diurnal centroid could give some information on changes in plant water recharge. Furthermore, the diurnal centroid base metrics complement the hysteresis quantification methods such as those employed by Zhou et al. [54] and Matheny et al. [29], with the advantage of $C_{ET}^*$ being compensated for cloudy conditions and possibly comparatively less influence of noise, though an intercomparison

would be useful to explore the strengths and weaknesses of the different approaches. By providing both the equations and related code of the metrics, we the authors hope the metrics will be used by the community for both validation and to further ecophysiological understanding.

**Trees, grass, and drought stress**

By comparing climate groups and PFTs with the frequent occurrence of low DWCI and $C_{ET}^*$ from Figure 4, we can note two striking differences: evergreen broad- and needle-leaf forests show high variability of morning shifted days but not uncoupled days, whereas grasslands show significantly high uncoupled but not morning shifted days. The pattern is further seen in Figure 5, where the distinct divergent responses of decoupling and morning shifts between tree and grass dominated systems. This disparity may indicate an interaction of $C_{ET}^*$ not only with drought, but hydraulic sensitivity. The association of morning shifts to hydraulic sensitivity is further strengthened by Figure 6a,d where $C_{ET}^*$ shows a much stronger response to EF rather than VPD, indicating that morning shifts of ET are not simply due to stomatal closure due to VPD but in fact a response to drought conditions. The shorter hydraulic system of grasses may not necessitate stomatal closure under high demands [14], thus causing less frequent phase shifts even under drought conditions. In contrast, tree ecosystems may only exhibit higher hydraulic stresses, associated with both dryness and a more sensitive hydraulic strategy. Temperate-continental and tropical climates all showed a low frequency of morning shifted days, even though they are occupied by large trees with cavitation susceptible vascular systems [21], suggesting that these ecosystems show limited drought stress even with the hydraulic susceptibility. Similarly, the high degree of variability of morning shifted frequency between site-years in sub-tropical/Mediterranean and evergreen broad- and needle-leaf forests could either indicate variation in the response in hydraulic stress between sites, or that hydraulic stress is only expressed some years, leading to high and low frequencies within the same site.

In this way, it seems that though $C_{ET}^*$ is less noisy as a drought indicator (see Figure 6), it may only be of use in tree systems that are more prone to hydraulic stress. However, this does put the metric in a rather unique position in that it could be used as a global scale hydraulic indicator, having potential application in exploring ecosystem level isohydricity [27], or the degree to which risks vascular system damage to continue to extract water. Isohydricity is intrinsically a concept that relates to an individual plant, as dynamics of rooting depth, hydraulic conductances, and sensitivities to VPD can vary within individuals of the same species at the same location. However, these factors are all interrelated, as hydraulic and stomatal conductances drive transpiration dynamics which control the rate of depletion of root zone water which can then feed back to stomatal sensitivity, such as via ABA signaling [49]. As such, current estimates of isohydricity require plant level measurements, which are currently restrained to the individual scale, i.e. from actual leaf measurements [28] or to global scale, but only 0.5 degree resolution estimates from radar [21]. This limitation of large and small scales leaves a knowledge gap at the size of an eddy covariance footprint, hindering the study of ecosystem

response to drought. However, under the assumption that the morning shifts seen under low evaporative fraction are related to increased stomatal sensitivity in response to root zone moisture depletion, it may be possible to compare the onset and speed with which the diurnal centroid shifts toward the mornings as ecosystems dry. In this way, one could infer the ecosystem response to soil moisture, without explicitly knowing the soil moisture. The resulting relationship could prove useful as a data derived ecosystem functional property, giving direct information on variations in water limitation response.

**C:W decoupling and energy balance closure**

In addition to error from single data points, both metrics, but especially the DWCI, show some relationship with energy balance over closure. Energy balance mismatch is a common phenomenon in EC measurements, with under closure (ET+H<Rn) being a more common concern [24, 51]. Issues with energy balance closure can be, among other causes, attributed to advection, where energy, water, and carbon are transported in and out of the tower footprint, complicating an absolute accounting of these quantities [1, 5, 50]. The apparent association of DWCI and over closure could be due to transfer of moist air from the surrounding landscape, causing the DWCI to be more contingent on the mixing of source air and less from plant controls. In this scheme, the over closure seen in Figure 6 could be caused by the mixing of outside moist air into the drier air from the EC site, causing an increase in latent energy. However, the infiltrating air sources could also have similar or drier moisture levels which would not necessarily be seen as over closure. In this scenario, this infiltrating air could contain varying carbon and water concentrations, again causing a carbon:water decoupling, but one that would not be associated with over closure. If this effect has no diurnal pattern, and thus does not generally influence the mean diurnal centroid in ET, it could explain why the patterns with dryness are much clearer with $C_{ET}^*$ compared to DWCI. This would have the implication that DWCI is then a mixture of advection and non-stomatal signals, complicating the biological interpretability. However, the association with dryness in both metrics gives credence that they do indeed reflect some physiology, if we assume EBC should not be influenced by dryness level. Furthermore, if potential stress conditions are removed, the DWCI could be useful as a metric of advection in the system, even when the energy balance is relatively well closed.

**WUE shifts associated with metrics and not captured by models**

Figure 7 demonstrates the strong tendency of the models to underestimation WUE in dry conditions. This is true even for the fully non-linear and empirical random forest model, indicating that the model under-performance is not necessarily due to an incomplete model framework, but due to a lack of information to constrain the problem. Given the association of both metrics with drought (Figure 6), one could expect that the models would underestimate WUE in uncoupled and

morning shifted days. Though this is the case with morning shifted days, decoupling shows no underestimations of WUE, with even a mean overestimation in the case of the Katul-Zhou and Boese models. Given the limitations outlined in the previous sections, one could blame noise for the lack of WUE shift, but this does not reconcile with the higher frequency of decoupling during dry days which should bias the WUE estimates. Furthermore, as the more empirical random forest model reduces the prediction variability, leaving a slight WUE underestimation, indicating that some of the overestimation from the Katul-Zhou and Boese models may be tied to limitations of the underlying assumptions, yet the distribution from the RF model still lacks the long tails of underestimation characteristic of the dry points. Extending these findings to the underlying hypotheses of the metrics, namely hydraulic and non-stomatal limitations, we could conclude that the hydraulic controls do impose a greater water use advantage than non-stomatal limitations. In other words, the findings suggest that days with water:carbon decoupling, and possibly non-stomatal limitations, do not improve WUE, whereas hydraulic responses can improve WUE. As WUE is a ratio, this does not shed any light onto the change in productivity, as low values of WUE may indicate that a plant is still productive, but at a higher water cost. However, solid conclusions would require further analysis with some site specifics measurements of actual plant function.

Though the models used here are relatively simple and lack the complexities and feedbacks found in more vigorous ecosystem models, Matheny et al. [29] also demonstrated the fundamental inability of 9 different land-surface models with 4 different stomatal conductance schemes to capture diurnal variability which the authors attributed to inadequate representation of how water gets from the soil to the leaf. Given the demonstrated phenomenon of morning shifts and decoupling across sites under dry conditions, the metrics here provide a benchmarking tool for mechanistic models to test their ability to replicate these patterns, suggesting that the models are capable of expressing hydraulic and non-stomatal limitations. Furthermore, in the case of machine learning approaches, the metrics may provide a useful input parameter which summarizes these diurnal effects, as is evidence by the difference in response the bias in RF modeled WUE, i.e. while both metrics are associated with low EF, RF WUE was underestimated with morning shifted days but not decoupled days implying that two different strategies are being captured by the metrics. As such, by demonstrating the utility of the metrics, and providing code and explanations for calculation, we hope they become useful to the community at large.

**Conclusion**

Both the DWCI and the $C_{ET}^*$ demonstrate an ability to show consistent patterns with drought across a broad array of sites, climates, and ecosystems, with the added advantage of being tied to theoretical underpinnings. Particularly, the demonstrated patterns give novel information about carbon water relations and hydrological dynamics that are not currently present at ecosystem scale across a database as large as FLUXNET. These metrics and their underlying theory provide a data derived example differentiating the hydrological response of tree and grass plant functional types, as well

as give evidence for the presence and absence of a WUE advantage from hydraulic and stomatal limitations respectively.
Going forward, these metrics can be used as a tool to further understand the diversity of ecosystem drought responses.

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
