# Peer review of "Water stress induced breakdown of carbon-water relations: indicators from diurnal FLUXNET patterns"

_Biogeosciences, 2017_

## Referee Comment (RC1) · M. G. De Kauwe (Referee) · 6 Nov 2017

Nelson et al. present a novel exploration of FLUXNET data to derive two new data-driven drought metrics. I think their approach is very interesting and this could form part of a more nuanced way to benchmark land models in the future. Ultimately, whilst I feel that this paper should be published, I think the text requires quite a lot of clarification and redrafting.

I found the introduction & methods text a bit disjointed. There is very little in the way of text to explain or set the "why is drought a big problem" argument. To me, there was a strange focus on considering the proportion of transpiration to ET. The authors spoke a

lot about uncertainty, but it wasn't very clear if their focus was data (flux measurements) or model world. In short, the authors could do a better job of framing the broader problem before they get to the hydraulic and non-stomatal limitation text. In the methods, you often jump or introduce new concepts with little back story and it becomes quite distracting.

I also think the analysis of the results could be more incisive. I strongly feel the authors are doing themselves a disservice in terms of likely citations by not picking through their results a bit further (see comments).

Finally, whilst I shouldn't have to, it still true that it isn't the norm to share code, so I applaud the authors. I suspect it is likely to lead to their work being more widely used and potentially improved upon.

Introduction ————

- I don't immediately follow why you've introduced VPD into equation 2? Surely your estimation of GPP and ET have both already accounted explicitly for a VPD dependence? Then on line Pg 1, line 19 you say "more consistent" ... more consistent with what? This is probably simply my ignorance, but I'd like to follow the logic here because you then use VPD in eqn 7 and 8.

- Pg 2, line 2: "propagate errors ..." I assume you mean in terms of a model? As actually measured fluxes would account for any drought signature? Please clarify.

- Pg 2: The arguments about the uncertainty of T as a proportion of ET ... do we really think that this the chief uncertainty here is drought? To me it feels like an odd framing of the argument simply because I wouldn't expect water stress to dominate the water cycle and the uncertainty range quoted is large. I think this line of argument would be improved by simply talking about the need to understand the carbon and water cycles during water stress. I'd argue for removing all of this text.

Methods ———-

- I think it would be helpful to explain why PET was calculated. The text just jumps to we calculated PET...Also, there is a brief mention of why the approach was adopted, but it should be expanded upon. Similarly with the CSWI, you just suddenly jump to explaining it without any back story for the reader.

- The screening of data to remove contributions from the soil is potentially problematic. I've seen that other authors have used 48 time slots after rain (see Medlyn et al 2017, New Phytologist and references they cite). The authors have taken a different approach, but screening GPP < 5 g m-2 d-1 seems quite high? Presumably as you get a drought, GPP drops and this may remove some of the signal you seek to explore? Similarly an air temp of 15 deg C. Whilst admittedly not "warm", doesn't it depend where you are? There are many locations with variable day-to-day temp, even in summer. Did the authors explore any sensitivities to these assumptions?

- Similarly the assumption about precipitation and gap filling. What about filling it with reanalysis data? Assuming a gap corresponds to a 5 mm precipitation event strikes me as quite a big assumption? What happens if you simply assumed a gap = no rain? How important is this assumption for your results?

- Page 5, line 24. Is there any evidence of this shift? I'm not arguing it isn't true, but the authors don't cite any supporting literature. Later on in the text the authors cite Wilson, but are there any other citations? It would be good to support this point. Figure 1 is nice and useful for demonstrating the authors point.

- Page 8: similarly to where I've made this point before, you really need to introduce things better. Suddenly the text jumps to the "Katul" and then the "Boese" models, with little or no back story. To this point I've found this paper really interesting, but these jumps honestly make it hard to follow and are quite distracting, so I hope I'm being constructive here.

- I don't really follow the benchmarking models? As to get ET, you use GPP derived from flux data and then measured VPD and Rg? Why do you need a benchmarking
model? To me, you simply need to apply the method to the raw data?

Results ——-

- Why when DWCI < 10 is it reasonable that you have decoupling? That is stated as a fact without any support? Ditto CET < -0.5. Can you not demonstrate this for a case study i.e. the 2003 summer data from Europe, or similar?

- The 7 to 8% of all points being decoupled at all sites. Does that make sense? Wouldn't you hypothesise differences based on the vegetation? Rather than expect to find a universal value? I realise you have large uncertainty bounds, but I wonder what the implication of that finding is? Does it imply anything about the method at all? I don't have an immediate suggestion, I'm simply surprised.

- It might be interesting to see figure 3 expressed in a more informative way. Perhaps by mean annual precipitation, or spring/summer precipitation and/or an aridity index? It would also be interesting to see how variable individual years are? You clearly have this information, but it is compressed in your presentation of Fig 3 and arguably this information is very interesting and I'd argue that you're selling your paper short by not exploring it. For example, how variable was 2003/2010 vs other years for European sites?

- Similarly, do you see a shift in the centroid related to specific times in the year? Which sites shift earlier? What physically can you tie this to?

- I don't find figure 4 all that informative. Again I wonder if you are exploiting the interesting findings to their fullest? Which sites are most decoupled? Which vegetation types? Does it make sense to exclude the well coupled days, you're not really interested in these days?

- I won't really comment on Fig 5 because I don't follow the motivation. Partly because of my question about VPD and partly because I don't see why the metrics which are data driven, require a benchmark like this? I'm not totally sold on this being an objective

means to test the approach, but appreciate why the authors have taken this approach. This is simply my opinion and I'm sure others would disagree. My first point of my discussion text below would be the way I would have been tempted to proceed.

Discussion ————-

- How do we know the method works? What would be the best test of the method? Even if the authors don't have access to the necessary data, could they set a challenge to the community? For example, if groups had sapflux or information on non-stomatal limitations at any flux sites, do the authors have thoughts how these data could be used? How should the community push such an approach forward?

- I think the discussion of trees vs grasses is interesting and welcomed, but I wonder if the authors looked at exploring a bit more within a functional group (i.e. by aridity etc), whether they might find something else too. Up to the authors of course.

- I'd argue that the authors could set aside some text to suggesting how their approach could be used in terms of benchmarking land models during drought? I'm not saying this paper has to do such a comparison, but it might be advantageous to lay the ground work. I'm guessing that the authors see modellers as potential users of their metrics? and if so, it is worth them making a case. Assessing models for responses to drought is very complicated and so their approach is welcomed.

---

## Referee Comment (RC2) · Anonymous Referee #3 · 7 Nov 2017

Overall, I am supportive of the goals of this study. I agree that the asymmetry of the diurnal cycle of ET, and the correlation between GPP and ET, likely contain meaningful information about ecosystem response to drought. I also applaud the author's efforts to link these metrics to insights informed by mechanistic theory.

The analysis presented here is very broad (i.e. results are synthesized across many sites, and often many PFTs). Cross-site syntheses like this are invaluable for understanding broad patterns in vegetation functioning; however, the objectives of this study would benefit from a more in-depth analysis of the results at a least few sites, preferably sites where there exist independent estimates of plant water relations during periods of hydrologic stress (i.e. from gas exchange, sap flux, isotope analysis of tree cores, etc). This sort of analysis would give the readers confidence that the C_ET and DWCI metrics are really reflecting stomatal and non-stomatal limitations to transpiration and carbon uptake, and aren't unduly contaminated by the many sources of uncertainty in using eddy covariance measurements to infer GPP and water use efficiency (e.g. Knauer et al. 2017). Or, to put it differently the new metrics introduced in this manuscript merit some "proof of concept" before they are applied broadly.

I also had a few concerns about the presentation and interpretation of the water use efficiency theory. First, the authors attributed the afternoon decline in ET to "hydraulic limitation" driven specifically by challenges of moving liquid water from roots to the leaves as soil dries (e.g. Lines 24-25). While I agree that hydraulics are an important control on stomatal functioning, stomates may also close directly in response to rising VPD even if soil moisture is unchanged (as discussed at length in the stomatal optimization literature), and the mechanisms responsible for the VPD response are still not yet clear. Thus, it may be more appropriate to describe the afternoon decline in ET as simply "stomatal limitations."

Next, the function ET=i*GPP*sqrt (VPD), proposed by Zhou et al. (2015), is referred to in this manuscript as the "Katul" model; presumably this nomenclature originates from the theory presented in Katul et al. (2010), which presents arguments leading to the equation:

ET = GPP x sqrt(VPD) / sqrt(1.6*lambda*ca)

The parameter lambda is the so-called "marginal water use efficiency" and ca is atmospheric CO2 concentration. This is similar to the Eq. 8 in the present manuscript:

ET = i x GPP x sqrt(VPD)

If i = 1/ sqrt(1.6*lambda*ca)

I appreciate that the authors have attributed the model to Katul et al. (2010), who

presented the theory on which the equation is based. However, before the authors attribute the model's "inability to make accurate predictions" to "be a result of a failure of their underlying assumptions," (Page 9, lines 10-11), care should be taken to make sure the underlying assumptions are properly stated and considered. The Katul et al. (2010) result relies on an assumption of Rubisco-limited photosynthesis (which generates a linear A-Ci curve)…this assumption is not likely to hold in dense forests where understory vegetation is often light-limited. Second, and perhaps more importantly, stomatal optimization theory assumes that the parameter lambda should hold constant over timescales of hours, but varies over longer timescales (days to weeks) as other slowly-evolving boundary conditions change (Manzoni et al. 2013, Palmroth et al. 2013). So in that regard, I disagree with the author's assessment on Page 8, Line 5, that the "Katul" model "makes the assumption that the WUE is constant if corrected by the effect of VPD." The potential for lambda to vary may also help to explain the tendency of the Katul (and other) models to underestimate WUE during dry conditions (i.e. Figure 5), especially if the 'i' parameter is determined using observations from well-watered conditions.

Finally, when using the shape of the diurnal pattern of ET to infer stomatal limitation, I wondered why the authors focused only on the shift in the peak, and not the overall degree of asymmetery between morning and afternoon periods (for example, if the ET data from hours 0-12 are reflected about the solar noon axis, that is the area between the reflected and actual ET data).

A few other comments include:

Page 2, lines 15-24. The presentation of the iWUE models would benefit from a more general explanation of what each of the three metrics actually describes and/or is sensitive too. E.g. WUE is useful for understanding broad patterns of ecosystem water use and carbon uptake, but is sensitive to non-biological drivers (e.g. VPD). The iWUE attempts to correct for the direct effect of VPD on transpiration, and is thus a more biologically relevant metric. The uWUE further attempts to correct for stomatal clo-

sure under high VPD, and therefore may be more closely linked to the "non-stomatal" limitations to gas exchange during drought.

Page 4, Line 17: "the use of EC measured diurnal patterns of carbon, water, and energy fluxes to derive clues on ecosystem drought responses at a daily resolution could prove valuable, if nothing less than a benchmark to test current hypotheses." As far as rationale for the work goes, I found this to be rather weak. Perhaps the authors could give specific examples of hypotheses that could be tested with these metrics.

Page 9, line 20: "sites under water stress tended to have $C\_ET < -0.50$." How do the authors know that the sites were under water stress? This gets back to my original point about validating the metrics against independent observations of plant function.

Figure 3: The text is small and hard to read.

Page 13, Lines 15-25: I found this discussion of the links between $C\_ET$ and isohydricity to be highly speculative, notably because isohydricity tends to describe plant response to declining soil moisture, yet the afternoon stomatal closure may be largely caused by increasing VPD.

References: Katul, G, Manzoni, S., Palmroth, S., Oren, R. 2010. A stomatal optimization theory to describe the effects of atmospheric CO2 on leaf photosynthesis and transpiration. Annals of Botany, 105, 431-442.

Knauer, J., Zaehle, S., Medlyn, B.E., Reichstein, M., Williams, C.A., Migliavacca, M., De Kauwe, M.G., Werner, C., Keitel, C., Kolari, P., Limousin, J.-M., Linderson, M.-L. 2017. Towards physiologically meaningful water-use efficiency estimates from eddy covariance data. Global Change Biology; doi:10.111/gcb.13893.

Manzoni, S., Vico, G., Porporato, A., Palmroth, S., Katul, G. 2010. Optimization of stomatal conductance for maximum carbon gain under dynamic soil moisture. Advances in Water Resources, 62, 90-105. Palmroth, S., Katul, G., Maier, C.A., Ward, E., Manzoni, S., Vico, G. 2013. On the complementary relationship between marginal

nitrogen and water-use efficiencies among Pinus taeda leaves grown under ambient and CO2-enriched environments. Annals of Botany, 111, 447-467.

---

## Referee Comment (RC3) · Anonymous Referee #2 · 14 Nov 2017

This study proposes two data-driven indicators using eddy covariance data to examine water stress induced breakdown of carbon-water relations. These results are scientifically interesting. Sharing code is a also good practice and should be praised, although the calculations seem to be straightforward. Given the problems with the manuscript, I recommend that it be reconsidered after a major revision.

The biggest problem of this manuscript is that the Results section is very weak. It only contains three relatively short paragraphs, which is far from being sufficient for a research article. The authors should substantially strengthen this section.

The authors claimed that they proposed two indicators. Are these indicators new and

have not been used in the literature? If they have been used in the literature, then they are not new and the authors should rephrase this statement. Take diurnal centroid as an example, it was used back in 2003 (Wilson et al. 2003), and what is new with the relative diurnal centroid?

The abstract should be rewritten to contain less introduction and more results.

The Introduction section is organized by sub-sections. The three levels of organization is pretty unusual for scientific papers. I suggest that the authors remove sub-section titles and rewrite it as a regular Introduction. If the authors intend to provide more background material, it is better to add a Background section right after the Introduction.

It is unclear which FLUXNET database (LaThuile or 2015) is used in this study. Details should be provided. The Baldocchi et al. 2008 paper does not seem to be the proper citation for the database used.

The supplementary figure can be moved to the manuscript given the relatively small number of illustrations.

Line 30, page 4: FileS1 is a nice way to present study sites. Adding some technical details about how this kind of file was created will be helpful to the audience who might want to produce this kind of illustration as well.

Figure 2: An overall title should be added for the figure. Moreover, "upper" and "lower" should be changed to something like "Upper panels:" and "Lower panels:", respectively.

Line 8, page 3: "are GPP" should be changed to "GPP are".

Figure 2: An overall title should be added for the figure. Moreover, "upper" and "lower" should be changed to something like "Upper panels:" and "Lower panels:", respectively.

Line 8, page 3: "are GPP" should be changed to "GPP are".

Line 25, page 8: remove "be".

---

## Author Comment (AC1) · 23 Jan 2018

**Response to interactive comment by M. G. De Kauwe (Referee) on "Water stress induced breakdown of carbon-water relations: indicators from diurnal FLUXNET patterns"**

Nelson et al. present a novel exploration of FLUXNET data to derive two new data-driven drought metrics. I think their approach is very interesting and this could form part of a more nuanced way to benchmark land models in the future. Ultimately, whilst I feel that this paper should be published, I think the text requires quite a lot of clarification and redrafting.

I found the introduction & methods text a bit disjointed. There is very little in the way of text to explain or set the "why is drought a big problem" argument. To me, there was a strange focus on considering the proportion of transpiration to ET. The authors spoke a lot about uncertainty, but it wasn't very clear if their focus was data (flux measurements) or model world. In short, the authors could do a better job of framing the broader problem before they get to the hydraulic and non-stomatal limitation text. In the methods, you often jump or introduce new concepts with little back story and it becomes quite distracting.

I also think the analysis of the results could be more incisive. I strongly feel the authors are doing themselves a disservice in terms of likely citations by not picking through their results a bit further (see comments).

Finally, whilst I shouldn't have to, it still true that it isn't the norm to share code, so I applaud the authors. I suspect it is likely to lead to their work being more widely used and potentially improved upon.

> We would like to thank Dr. De Kauwe for this very in-depth and thorough review. As authors it is always welcome to get such thoughtful outside opinions to further strengthen and clarify a manuscript. We hope the revised manuscript has a much better flow which allows the reader to follow our line of thinking and limit the distractions, as well as the expanded analysis and discussion. Following are the detailed responses which we hope address these overall concerns.

Introduction ————

1. I don't immediately follow why you've introduced VPD into equation 2? Surely your estimation of GPP and ET have both already accounted explicitly for a VPD dependence? Then on line Pg 1, line 19 you say "more consistent" ... more consistent with what? This is probably simply my ignorance, but I'd like to follow the logic here because you then use VPD in eqn 7 and 8.

> We agree with the reviewer that this concept may have been inadequately introduced. It may have been rather distracting to introduce both the uWUE and iWUE equations, and as only the uWUE was use in the calculations of the DWCI and the Katul model. As such, the iWUE references and equations have been eliminated. Furthermore, we have tried to highlight that the $\sqrt{VPD}$ allows for a more stable metric which can be compared across timescales, which becomes important when comparing diurnal GPP and

ET in calculating the DWCI. Therefore we have rewritten this section, which also incorporates comment 3 from reviewer 3. This subsection now reads:

Classically, vegetation water and carbon fluxes are linked by stomates, where an open stomate allows CO2 to enter the leaf and, consequentially, water is lost. From this, most theoretical frameworks make some form of assumption that carbon assimilation (A) and water losses (T) are both contingent primarily on leaf stomatal conductance (gs). This assumed relationship allows us to pass between the realms of carbon and water, based on the assumption that at any given time both A and T are proportional to the stomatal conductance multiplied by the difference in internal and external CO2 and water vapor concentrations. More specifically,

$$A = g_s \cdot \Delta c \quad and \quad T = 1.6 \cdot g_s \cdot \Delta v$$

where Δc and Δv are the differences in inner and outer stomatal cavity concentrations of CO2 and water vapor, respectively. These diffusion equations lead to the relatively consistent carbon:water ratio, generally expressed as a water use efficiency ($WUE = A/T$). At the ecosystem level where direct measurements of A and T are not available, WUE is simply calculated as the ratio of gross primary productivity (GPP) to total evapotranspiration (ET) [7]. These carbon:water links are fundamental to understanding how stomata are regulated and underly key functioning in mechanistic plant and ecosystem models. One such set of models are those based on optimality theory which posit that plants tend to optimize carbon gains to water losses, such as those described by Katul et al. [5] and Katul, Palmroth, and Oren [4]. These concepts from Katul, which carry the assumptions of RuBISCO (light) limitaion, were built upon by Zhou et al. [14] and Zhou et al. [13] to give the equation,

$$uWUE = \frac{GPP \cdot \sqrt{VPD}}{ET}$$

where the $\sqrt{VPD}$ accounts for the stomatal response to vapor pressure deficit (VPD). Accounting for the VPD response allows for a more stable metric of WUE that is temporally more stable and physiologically more meaningful, such as when comparing the diurnal cycles of carbon and water. As ET is the sum of both T and non-biological evaportaion (e.g. soil and intercepted evaporation), often periods during and shortly after rain events are excluded from WUE estimates to minimize the influence of non-plant evaporation. Ultimately, calculations of WUE provide a simple summary of the cost in water per carbon gain and becomes an indicator for how plants have and will adapt to the physical limitations of their changing environments [6, 11].

2. Pg 2, line 2: "propagate errors ..." I assume you mean in terms of a model? As actually measured fluxes would account for any drought signature? Please clarify.

> Indeed this wording was unnecessarily vague. Here we were referring to assumptions on carbon and water in the models which then affects carbon and water estimates. To clarify this point the text has been changed to:

>> This failure to capture the effects of drought is not only disconcerting as water limited conditions are when ecosystems are most at risk, but an incomplete framework tends to propagates errors and uncertainties from models into estimates of the water and carbon cycles.

3. Pg 2: The arguments about the uncertainty of T as a proportion of ET ... do we really think that this the chief uncertainty here is drought? To me it feels like an odd framing of the argument simply because I wouldn't expect water stress to dominate the water cycle and the uncertainty range quoted is large. I think this line of argument would be improved by simply talking about the need to understand the carbon and water cycles during water stress. I'd argue for removing all of this text.

> This section came in part from the fact that this research was motivated from a transpiration angle, and also to highlight that gaps in understanding of drought responses can be reflected in both the water and carbon cycle communities. However, as other reviewers have also noted that the introduction could be shortened, and this section has maybe become less relevant, it has been removed.

Methods ——-

4. I think it would be helpful to explain why PET was calculated. The text just jumps to we calculated PET...Also, there is a brief mention of why the approach was adopted, but it should be expanded upon. Similarly with the CSWI, you just suddenly jump to explaining it without any back story for the reader.

> We agree with the reviewer that the reasoning for calculating these parameters was not made clear, instead we simply jumped into the calculations. The PET was used in calculating evaporative fraction, which gives a consistent measure of ecosystem dryness across sites with which to compare the metrics (basically a daily aridity index). This concept is now introduced in the text as:

>> In order to provide a consistent measure of ecosystem dryness that can be utilized across sites, the ratio of water evaporated to potential water evaporated was calculated as evaporative fraction (EF), or the fraction of actual ET to Potential ET (PET).

> Likewise, the CSWI was used as a replacement to the time after rain methods for screening periods with wet surfaces and likely higher evaporation. We now mention this when introducing the concept:

>> In an effort to minimize contributions of evaporation, the conservative soil wetness index

(CSWI) was employed which was designed to estimate whether the ecosystem is likely to have "dry" surfaces and therefore ET is likely to be dominated by transpiration. This approach requires a certain amount of evaporation to occur after a rain event before the surface is considered to be "dry" and can be contrasted to the method of removing a set time period after rain employed in previous studies [9, 1, 6].

Hopefully by introducing these topics before jumping into the details of their calculation this section with flow better and not be so jarring.

5. The screening of data to remove contributions from the soil is potentially problematic. I've seen that other authors have used 48 time slots after rain (see Medlyn et al 2017, New Phytologist and references they cite). The authors have taken a different approach, but screening GPP < 5 g m-2 d-1 seems quite high? Presumably as you get a drought, GPP drops and this may remove some of the signal you seek to explore? Similarly an air temp of 15 deg C. Whilst admittedly not "warm", doesn't it depend where you are? There are many locations with variable day-to-day temp, even in summer. Did the authors explore any sensitivities to these assumptions?

> The reviewer makes a good point that these thresholds deserve further examination. As outlined in the previous response, the CSWI was designed to be an improvement to the time after rain method, as this does not account for differences in evaporation, say if there is little radiation in the 48 hours after the rain event. However, they are difficult to compare, as they are not immediatly comparable. Upon further analyis, the median time period for the CSWI to go from fully wet (CSWI=5) to "dry" (CSWI<=0) was 3.5 days in summer (summer being periods when daily potential radiation above median daily potential radiation for that site), so more conservative than 1-2 days after rain, but on the same order of magnitude. This comparison has been added to the manuscript:

>> This method was used over the more standard method of removing 1-5 days after a rain event, as it does not make the assumption that the surface will dry in a fixed amount of time, instead relying on a minimum amount of evaporation. As a comparison, the median time period for the CSWI to go from fully wet (CSWI=5) to "dry" (CSWI<=0) was 3.5 days across all sites in summer, where summer was defined as the period when daily potential radiation above median daily potential radiation for each site.

> As to the GPP and air temperature limits, we have added a sensitivity analysis showing the response of DWCI and $C_{LE}^*$ to EF within various ranges of GPP and air temperature, Figure S2 (here R1). Based on this analysis, we looked for thresholds which would show a strong signal in the metrics under increasingly dry conditions (lower EF). In regards to temperature, our initial threshold does seem to be a bit high, as temperature thresholds above 5° C showed good metric response with reasonable variability under unstressed conditions particularly with the DWCI. As to GPP, is seems we had a problem with the units

[Figure]

Figure R 1: Figure S1. Sensitivity analysis of DWCI and $C^*_{ET}$ to evaporative fraction (EF=ET/potential ET) under different GPP and air temperature (Tair) values, as well as the sensitivity of frequency of decoupling (DWCI < threshold) and morning ($C^*_{ET}$ < threshold) using various thresholds. Note DWCI of 0-100 indicate lowest-highest probability of diurnal carbon:water coupling and $C^*_{ET}$ of -1-1 indicate one hour morning shifted-one hour afternoon shifted ET. Bins for GPP and Tair based on percentiles to give an equal number of points per bin, with the exception of GPP<1 and Tair<0, which are approximately double the size of the other bins and represent periods of low plant activity such as winter. Vertical bars represent interquartile range in all cases.

where the threshold was reported in grams $CO_2 \cdot m^{-2} \cdot d^{-1}$ instead of $gC \cdot m^{-2} \cdot d^{-1}$. With the unit conversion, this would correspond to a threshold of 1.36 $gC \cdot m^{-2} \cdot d^{-1}$. We would like to thank the reviewer for spotting the error which has been corrected in the text as well in the figures. Based on the sensitivity analysis in figure S2 (R1), we found that a threshold of 1 $gC \cdot m^{-2} \cdot d^{-1}$ works well in minimizing unnecessary variation in the metrics, as evidence by the smaller variability in the metrics during wet conditions. The issue is now discussed in the Data section in the Methods and Materials:

> Growing season was defined as all days where GPP > 1 $gC \cdot m^{-2} \cdot d^{-1}$ and daily mean air
> temperature > 5 ºC. These threshold were shown to give good response in the proposed metrics
> while minimizing variability due to low diurnal signals, a sensitivity analysis of which can be
> found in supplementary Figure S2.

6. Similarly the assumption about precipitation and gap filling. What about filling it with reanalysis data? Assuming a gap corresponds to a 5 mm precipitation event strikes me as quite a big assumption? What happens if you simply assumed a gap = no rain? How important is this assumption for your results?

> As precipitation data can be quite hard to predict, and we did not have immediate access to the reanalysis
> data for this dataset, we found it prudent to simply remove those days. In this way the data screening is
> conservative, erring on the side of caution by removing periods if we are not sure if it has rained or not.
> This conservative take is possible due to the large size of the dataset. Perhaps a further analysis with the
> FLUXNET 2015 database would yield more data points as some of these issues have been address in the
> new dataset, such as inclusion of reanalysis data gap-filling. However, to make this point clear in the
> manuscript, we have amended the text to read:

> > Any gaps in the precipitation data were assumed to be a precipitation event of 5 mm in order to
> > prevent any unmeasured precipitation event from biasing the results by inadvertently including
> > rainy days.

7. Page 5, line 24. Is there any evidence of this shift? I'm not arguing it isn't true, but the authors don't cite any supporting literature. Later on in the text the authors cite Wilson, but are there any other citations? It would be good to support this point. Figure 1 is nice and useful for demonstrating the authors point.

> Indeed there has been work on shifts in ET fluxes in the literature. We have added to this section to
> highlight some of the relevant work, as well as further highlight the work of Wilson and it now reads:

> > As soils dry, it becomes more difficult to transport stem and root zone moisture to the leaf,
> > potentially causing hydraulic limitations for the plant to transport water. This shift was seen in
> > eddy covariance data in a study by Wilson et al. [12], who examined the shift of latent compared
> > to sensible heat, which suggested that a shift in water fluxes towards dawn can be indicative

of afternoon stomatal closure. Shifts were further explored in a modeling study by Matheny et al. [8] which found that the morning shift was not well captured by models and attributed the errors to inadequate hydraulic limitations in the models. The daily cycle of wetting and drying acts as a capacitor in the hydraulic circuit, allowing water stores to be more easily transported in the morning and depleting in the afternoon. As bulk soil moisture declines, this effect may be strong enough to shift the diurnal cycle of ET significantly toward the morning. Quantifying diurnal shifts in EC data using the diurnal centroid was first explored by Wilson et al. [12]: defined as the flux weighted mean hour, or

$$C_{flux} = \frac{\sum flux_t \cdot t}{\sum flux_t}$$

where t is a regular, sub-daily time interval.

8. Page 8: similarly to where I've made this point before, you really need to introduce things better. Suddenly the text jumps to the "Katul" and then the "Boese" models, with little or no back story. To this point I've found this paper really interesting, but these jumps honestly make it hard to follow and are quite distracting, so I hope I'm being constructive here.

Please see the response to the next comment.

- I don't really follow the benchmarking models? As to get ET, you use GPP derived from flux data and then measured VPD and Rg? Why do you need a benchmarking model? To me, you simply need to apply the method to the raw data?

We find the reviewers comments very constructive and indicative of the high level of attention given to the review. We agree this has not been made clear and, as a consequence, has made subsequent parts of the manuscript unclear. The metrics were always calculated from the original flux data. The point of the model exercise was to benchmark whether the metrics were capturing information that the models were unable to predict, thus indicating that we are indeed providing novel information with these metrics and not something the models already capture. The idea is that if the models are unable to capture the variability of WUE, but the metrics are, then this indicates that the metrics are indeed providing some new and useful information that the models could benefit from. Here the Katul model would represent something closer to theory with some underlying assumptions, and the random forest would be at the opposite end of the spectrum, being wholly empirical with no assumptions. Therefore, if the metrics are capturing variability that the Katul model is not, it could indicate a problem with our assumptions in implementing the Katul model. However, if the metric captures variability that the random forest is not able to capture, it could indicate we are indeed adding new information that not even empirical methods are capturing. As this seems to be a key point that was causing confusion, in addition to the expanded section in the Introduction as discussed in comment 1, this section has been revised to outline

the motivation for using the models and now starts:

> In order to benchmark whether these metrics are capturing information that is possibly not being captured in modern model frameworks, three simple models were used to estimate WUE (GPP/ET) for each day at each site and compared to actual flux data. The purpose of the exercise was to evaluate if bias in the model predictions were associated with decoupled or morning shifted days, thus indicating that the metrics are corresponding to information that the models are unable to capture. Here we utilize three models to provide a spectrum of theoretical to empirical basis.

Results ——-

9. Why when DWCI < 10 is it reasonable that you have decoupling? That is stated as a fact without any support? Ditto CET < -0.5. Can you not demonstrate this for a case study i.e. the 2003 summer data from Europe, or similar?

> Based on the revieweres suggestion, we have added a case study of 6 sites during the 2003 heatwave, the results of which can be seen in the new Figure 3 (here R2). This new section shows that both monthly median values and interquartile range of DWCI and $C_{ET}^*$ did respond during the heat wave years, although the response varied both between the sites and in timing that sites showed a response. The results of this case study have been summarize in the manuscript with the following addition:

[revised manuscript text omitted]

10. The 7 to 8% of all points being decoupled at all sites. Does that make sense? Wouldn't you hypothesize differences based on the vegetation? Rather than expect to find a universal value? I realize you have large uncertainty bounds, but I wonder what the implication of that finding is? Does it imply anything about the method at all? I don't have an immediate suggestion, I'm simply surprised.

Indeed this statement was rather misleading, as we did not mean to indicate that 7-8% of points being decoupled at all sites. The number of points varied by site, as well as by climate and PFT as seen in Figure 3 (now Figure 4 in the updated manuscript). We have changed this section to hopefully clarify this point, as well and reference the map in File S1, which also gives information of the decoupling and morning shifts on a per site basis. However, we would again like to note that these thresholds are flexible, as the patterns are robust across thresholds from ranges of about 5-75 for DWCI and -1.0-0.0 for $C_{ET}^*$. As seen in the previous comment, these points have been added to the manuscript.

11. It might be interesting to see figure 3 expressed in a more informative way. Perhaps by mean annual precipitation, or spring/summer precipitation and/or an aridity index? It would also be interesting to see how variable individual years are? You clearly have this information, but it is compressed in your presentation of Fig 3 and arguably this information is very interesting and I'd argue that you're selling your paper short by not exploring it. For example, how variable was 2003/2010 vs other years for European sites?

The reviewer makes a good point that by beginning the results with a figure that was meant to give a broad overview across all 189 sites does obscure to the reader the seasonal and interannual dynamics. There-fore we have added the case study of the European sites (now Figure 3 (R2) in the updated manuscript) which shows both the regular seasonal cycles in the dry subterranean site of Puechabon forest, and the response of the metrics in 2003 across the European sites (as discussed in the response to comment 9). Furthermore, to focus on the differences in response of the metrics between tree and grass ecosystems, we have added Figure 5 (R3) which shows that tree ecosystems show a heightened response to diurnal centroid under lower evaporative fraction levels, while grasses and savannas responded more with the DWCI. This new analysis helps support the discussion on tree and grass responses (in the Discussion sec-tion). Hopefully this added analysis, along with the original Figures 3-4 (now Figures 4 and 6) will give the reader a better idea of how the metrics respond across environmental conditions and ecosystems.

12. Similarly, do you see a shift in the centroid related to specific times in the year? Which sites shift earlier? What physically can you tie this to?

[Figure]

Figure R 3: Median diurnal water carbon index (DWCI, upper panel) and diurnal centroid ($C^*_{ET}$, lower panel) of plant flunctional types binned by evaporative Fraction (EF, low values indicate dry conditions). Note DWCI of 0-100 indicate lowest-highest probability of diurnal carbon:water coupling and $C^*_{ET}$ of -1-1 indicate one hour morning shifted-one hour afternoon shifted ET. Evergreen needleleaf (ENF), deciduous broadleaf (DBF), and evergreen boradleaf (EBF) forests show increased morning shifts (low $C^*_{ET}$) with decreasing EF when compared to grassland (GRA) sites which tended to have decreased carbon:water decoupling (low DWCI) with decreasing EF. Savanna ecosystems (SAV) show a high degree of decoupling and intermediate levels of morning shifts. Vertical bars represent interquartile range.

As seen in the new Figure 3 (R2) on the European drought, some sites such as the Mediterranean oak forest Puechabon show a regular seasonal morning shift. More dramatic shifts can be seen at some sites during the 2003 drought, such as at Loobos forest in the Netherlands. The diversity of responses would hopefully indicate that the ecosystems are responding not just to atmospheric conditions but also soil water availability to the ecosystem, which we did not have the relevant data to properly analyse. However, Figure 4 (Figure 6 in the new manuscript, here R4) shows that the cleanest pattern in diurnal centroid is associated with periods of high net radiation and low latent energy, which is indicative water limited conditions. As for the DWCI, we see in the sensitivity analysis to air temperature (Figure S2 (R1)) that the decoupling tends to happen under wetter conditions at higher temperatures, so there may be a temperature component as well. We hope the added analysis focusing on particular sites during the European droughts give a better picture of how the metrics respond on different seasonal timescales.

13. I don't find figure 4 all that informative. Again I wonder if you are exploiting the interesting findings to their fullest? Which sites are most decoupled? Which vegetation types? Does it make sense to exclude the well coupled days, you're not really interested in these days?

Indeed as Figure 4 (here R4, now Figure 6 in the new manuscript) was intended to show that the response of the metrics could be seen across all sites, indicating that the metrics are indeed universally useful. However, this broad analysis then masks the dynamics seen from site to site and at different times of year. As mentioned in the previous comment, we see from the new Figure 5 (R3) that grassland and savanna ecosystems tend to decouple more and at higher evaporative fractions than tree ecosystems, possibly due to a lack of hydraulic architecture as discussed in the discussion sub-section "trees, grass, and drought stress". Regarding excluding the coupled days with regards to Figure 4 (R4, now Figure 6), as we are hoping to show the universality of the response, we want to be transparent and not mask any "false positives" one would see, as is the case of the noisiness of the response of DWCI in Figure 4a ((R4b) now Figure 6b). To highlight this point, the Figure is now referenced in the manuscript as:

The response of both variables to drought stress is further observed in Figure 6, where low mean values of both DWCI and $C_{ET}^*$ are associated with conditions of high net radiation and low latent energy, indicative of drought. As this figure includes all days from all sites which meet the filtering outlined in the Data subsection of the Methods, i.e. dry periods in the growing season, these figures exhibit the universality of the metrics across climates, ecosystems, and time periods.

Note the new subfigures (R4a,d), which were a response to the comments of reviewer 3 (comment 2)

14. I won't really comment on Fig 5 because I don't follow the motivation. Partly because of my question about VPD and partly because I don't see why the metrics which are data driven, require a benchmark like this? I'm not totally sold on

[Figure]

Figure R 4: Mean DWCI (upper panels) and $C^*_{ET}$ (lower panels) with respect to evaporative fraction (EF) by vapor pressure deficit VPD (a,d), latent energy (LE) by Rn (b,e) and LE by GPP (c,g). Note DWCI of 0-100 indicate lowest-highest probability of diurnal carbon:water coupling and $C^*_{ET}$ of -1-1 indicate one hour morning shifted-one hour afternoon shifted ET. Points with high Rn and low LE are associated with both low DWCI and $C^*_{ET}$, indicating that both metrics are related to water limitations. Though both metrics are associated with low EF, DWCI shows a much higher response to atmospheric demand as measured by VPD, with $C^*_{ET}$ showing very limited response. Both metrics, and DWCI in particular, show low values with high ET and low Rn, though these points are also associated with over closed energy balances (LE+H>Rn-G). Both metrics are associated with low GPP, but the $C^*_{ET}$ is restricted to both low GPP and ET, indicating water and carbon can decouple over a wider range of water stress. This also holds when points with energy balance over-closer are excluded (data not shown).

this being an objective means to test the approach, but appreciate why the authors have taken this approach. This is simply my opinion and I'm sure others would disagree. My first point of my discussion text below would be the way I would have been tempted to proceed.

> The idea behind Figure 5 (now Figure 7) was to really get at the question of whether the metrics are giving some new information on carbon water dynamics that a model couldn't capture, as discussed in the response to comment 8. As it is clear by the responses of all the reviewers, this point was not originally made clear to the reader. In an attempt to clarify this point, we have added text to highlight this point when outlining Figure 5(now 7) which also refers back to the expanded explanation in the materials and methods:

> > Figure 7 shows the difference between expected and observed WUE from the Katul, Boese, and random forest (RF) models, with respect to conditions of drought as characterized by low evaporative fraction (EF<0.2), C:W decoupling (DWCI<25), and morning shifts ($C^*_{ET}$<-0.25). This exercise was designed to test whether the metrics were associated with bias in the models, indicating that the metrics are able to capture information that the models are not (as further outlined in Methods and Materials subsection "models and parameter estimation").

Discussion ———-

15. How do we know the method works? What would be the best test of the method? Even if the authors don't have access to the necessary data, could they set a challenge to the community? For example, if groups had sapflux or information on non-stomatal limitations at any flux sites, do the authors have thoughts how these data could be used? How should the community push such an approach forward?

> We agree with the reviewer that though we have demonstrated the phenomenon, this broader approach would be supplemented by a site level investigation. With the added figures that show the response of the metrics during the heatwave of 2003, we hope that we give readers confidence that the metrics are indeed showing ecosystem stress. Furthermore, in an effort to make a call to the community we have added the following section to the discussion:

> > Given the broad nature of the analysis here, the metrics and hypothesis presented would benefit from site specific validations such as looking to see if the morning shits and decoupling are indeed associated with lower soil moisture levels, leaf water potentials, and/or decreases in sap flux. Sap flux in particular could give some interesting insights, as the diurnal patters in sap flux velocity will also have an offset to incoming radiation related to tree capacitance, therefore relating sap flow diurnal centroids to the ET diurnal centroid could give some information on changes in plant water recharge. Furthermore, the diurnal centroid base metrics complement

the hysteresis quantification methods such as those employed by Zhou et al. [14] and Matheny et al. [8], with the advantage of $C^*_{ET}$ being compensation for cloudy conditions and possibly less influence of noise, though an intercomparison would be useful to explore the strengths and weaknesses of the different approaches. By providing both the equations and related code of the metrics, we the authors hope the metrics will be used by the community for both validation and to further ecophysiological understanding.

15. I think the discussion of trees vs grasses is interesting and welcomed, but I wonder if the authors looked at exploring a bit more within a functional group (i.e. by aridity etc), whether they might find something else too. Up to the authors of course.

> Based on the reviewers suggestion, we have added Figure 5 (R3) which shows a more in depth view of the tree vs grass responses based on binned evaporative fraction. The figure further shows the distinct patterns with tree and grass responses, which we have highlighted in this section of the discussion.

16. I'd argue that the authors could set aside some text to suggesting how their approach could be used in terms of benchmarking land models during drought? I'm not saying this paper has to do such a comparison, but it might be advantageous to lay the ground work. I'm guessing that the authors see modellers as potential users of their metrics? and if so, it is worth them making a case. Assessing models for responses to drought is very complicated and so their approach is welcomed.

> We agree with the reviewer that this would be a useful addition and have added the following text to the discussion:

[revised manuscript text omitted]

---

## Author Comment (AC2) · 23 Jan 2018

**Response to interactive comment by Anonymous Referee #2 on "Water stress induced breakdown of carbon-water relations: indicators from diurnal FLUXNET patterns"**

This study proposes two data-driven indicators using eddy covariance data to examine water stress induced breakdown of carbon-water relations. These results are scientifically interesting. Sharing code is a also good practice and should be praised, although the calculations seem to be straightforward. Given the problems with the manuscript, I recommend that it be reconsidered after a major revision.

1. The biggest problem of this manuscript is that the Results section is very weak. It only contains three relatively short paragraphs, which is far from being sufficient for a research article. The authors should substantially strengthen this section.

    As all of the reviewers have made this point, we have expanded this section considerably. First we have included a new section focused on a case study of the European heatwave of 2003, in which one can see morning shifts and decoupling of water and carbon at 6 sites across Europe (now Figure 3 in the new manuscript, here Figure (R1)). The following text has been added to the results in relation to this figure:

[Figure]

Figure R 1: Monthly median diurnal water carbon index (DWCI, lower panels) and diurnal centroids ($C_{ET}^*$, upper panels) for 6 sites in Europe. Data from all years available (black) is compared to 2003 (red) during which a drought event resulted in high temperatures and low precipitation throughout the summer. Note DWCI of 0-100 indicate lowest-highest probability of diurnal carbon:water coupling and $C_{ET}^*$ of -1-1 indicate one hour morning shifted-one hour afternoon shifted ET. Vertical bars represent interquartile range. Sites from 5 plant functional types: evergreen broadleaf (EBF), deciduous broadleaf (DBF) and evergeen needleleaf (ENF) forests, as well as grasslands (GRA). Ecosystems show tendancies of morning shifts (e.g. DK-Sor and IT-Mal) and carbon:water decoupling (e.g. ES-ES1 and HU-Bug) during the drought year.

    As a case study, $C_{ET}^*$ and DWCI time-courses for eight sites from Europe are shown in Figure 3, with an emphasis on 2003 when the continent was struck by a heatwave that was shown to effect

both the carbon and water cycles [1, 4, 2]. For DWCI, forest sties showed high water:carbon coupling throughout the growing season, with the exception of Peuchebon (FR-Pue) which showed a regular seasonal cycle of decoupling. The grassland site (HU-Bg) showed a higher variability in DWCI compared to the forest sites (all others). All sites showed either a decrease in median DWCI or an increase in variability during 2003, generally in July or August, particularly at Hainich (DE-Hai), Bugacpuszta (HU-Bug), and El Saler (ES-ES1). This increase in decoupling during 2003 is consistent with the hypothesis of non-stomatal limitations being expressed in hot, dry conditions. Median diurnal centroid values across all years varied in absolute magnitude, but were generally near or above zero, i.e. the water cycle showed no shift or an afternoon shift. One exception would be the Mediterranean oak forest of Puechabon, which shows a slight seasonal cycle of morning shifts going from a slight afternoon shift to a slight morning shift during June, July, and August. During drought years, sites that showed distinctive morning shifts were Puechabon (FR-Pue), Soroe (DK-Sor), and Loobos (NL-Loo). The framework that morning shifts are associated with water stress from soil moisture depletion would be supported by the increase in morning shifts during 2003, though factors such as species composition and access to soil water would play a significant factor and could account for the differences among sites. All sites which had significantly different (p<0.05, Wilcoxon rank-sum test) DWCI values between 2003 and all other years except Puechabon, whereas with $C_{ET}^*$ only Puechabon, Soroe, and Loobos showed significant differences.

Furthermore, to highlight the divergent responses of the metrics between tree and grass dominated ecosystems, we have added a figure which shows the response of forest, savanna, and grassland $C_{ET}^*$ and DWCI values binned by evaporative fraction (now Figure 5, here R2). In the figure, one can see the tendency for grassland sites to decouple under low evaporative fraction, while forest sites tend to show morning shifts, which ties into the discussion on tree vs grass responses and isohydricity.

Finally, we have added two new subplots to Figure 3 (now Figure 6 in the new manuscript, here (R3)), which in addition to current subplots showing the DWCI and $C_{ET}^*$ response to LE by Rn and LE by GPP, shows the response to VPD by evaporative fraction (EF). These new subplots further differentiate the responses of the two metrics, as DWCI shows a combined effect of VPD and EF but $C_{ET}^*$ only shows a response to EF and not VPD.

We hope these new plots, as well as the associated references to them in the results, give the reader a better understanding of the dynamics of the metrics and further support our subsequent Discussion section.

2. The authors claimed that they proposed two indicators. Are these indicators new and have not been used in the literature?

[Figure]

Figure R 2: Median diurnal water carbon index (DWCI, upper panel) and diurnal centroid ($C^*_{ET}$, lower panel) of plant flunctional types binned by evaporative Fraction (EF, low values indicate dry conditions). Note DWCI of 0-100 indicate lowest-highest probability of diurnal carbon:water coupling and $C^*_{ET}$ of -1-1 indicate one hour morning shifted-one hour afternoon shifted ET. Evergreen needleleaf (ENF), deciduous broadleaf (DBF), and evergreen boradleaf (EBF) forests show increased morning shifts (low $C^*_{ET}$) with decreasing EF when compared to grassland (GRA) sites which tended to have decreased carbon:water decoupling (low DWCI) with decreasing EF. Savanna ecosystems (SAV) show a high degree of decoupling and intermediate levels of morning shifts. Vertical bars represent interquartile range.

[Figure]

Figure R 3: Mean DWCI (upper panels) and $C_{ET}^*$ (lower panels) with respect to evaporative fraction (EF) by vapor pressure deficit VPD (a,d), latent energy (LE) by Rn (b,e) and LE by GPP (c,g). Note DWCI of 0-100 indicate lowest-highest probability of diurnal carbon:water coupling and $C_{ET}^*$ of -1-1 indicate one hour morning shifted-one hour afternoon shifted ET. Points with high Rn and low LE are associated with both low DWCI and $C_{ET}^*$, indicating that both metrics are related to water limitations. Though both metrics are associated with low EF, DWCI shows a much higher response to atmospheric demand as measured by VPD, with $C_{ET}^*$ showing very limited response. Both metrics, and DWCI in particular, show low values with high ET and low Rn, though these points are also associated with over closed energy balances (LE+H>Rn-G). Both metrics are associated with low GPP, but the $C_{ET}^*$ is restricted to both low GPP and ET, indicating water and carbon can decouple over a wider range of water stress. This also holds when points with energy balance over-closer are excluded (data not shown).

If they have been used in the literature, then they are not new and the authors should rephrase this statement. Take diurnal centroid as an example, it was used back in 2003 (Wilson et al. 2003), and what is new with the relative diurnal centroid?

> The reviewer makes a good point that the relative diurnal centroid builds on the work of Wilson et al. [5] and should be properly credited. To this end the objectives have been amended to clarify that we propose these metrics particularly as indicators of water stress and now reads:

> > To this end, we propose two data-driven indicators of water stress, the diurnal water:carbon index (DWCI) and the relative diurnal centroid in LE ($C_{ET}^*$).

> We have also made the reference to Wilson et al. [5] more explicit, as well as highlighting the work of Matheny et al. [3] when introducing the metric:

> > As soils dry, it becomes more difficult to transport stem and root zone moisture to the leaf, potentially causing hydraulic limitations for the plant to transport water. This shift was seen in eddy covariance data in a study by Wilson et al. [5], who examined the shift of latent compared to sensible heat, which suggested that a shift in water fluxes towards dawn can be indicative of afternoon stomatal closure. Shifts were further explored in a modeling study by Matheny et al. [3] which found that the morning shift was not well captured by models and attributed the errors to inadequate hydraulic limitations in the models. The daily cycle of wetting and drying acts as a capacitor in the hydraulic circuit, allowing water stores to be more easily transported in the morning and depleting in the afternoon. As bulk soil moisture declines, this effect may be strong enough to shift the diurnal cycle of ET significantly toward the morning. Quantifying diurnal shifts in EC data using the diurnal centroid was first explored by Wilson et al. [5]: defined as the flux weighted mean hour, or

$$C_{flux} = \frac{\sum flux_t \cdot t}{\sum flux_t}$$

> > where t is a regular, sub-daily time interval.

> Hopefully these changes give better context to the manuscript and proper credit to previous works.

3. The abstract should be rewritten to contain less introduction and more results.

> The abstract has been rewritten to remove some introduction as well as to highlight the added analysis from the European heatwave in 2003:

> > Understanding of terrestrial carbon and water cycles is currently hampered by an uncertainty in how to capture the large variety of plant responses to drought. In FLUXNET, the global network of $CO_2$ and $H_2O$ flux observations, many sites do not uniformly report the ancillary variables

needed to study drought response physiology. To this end, we outline two data-driven indicators based on diurnal energy, water, and carbon flux patterns derived directly from the eddy covariance data and based on theorized physiological responses to hydraulic and non-stomatal limitations. Hydraulic limitations (i.e. intra-plant limitations to water movement) are proxied using the relative diurnal centroid ($C_{ET}^*$), which measures the degree to which the flux of evapotranspiration (ET) is shifted toward the morning. Non-stomatal limitations (e.g. inhibitions of biochemical reactions, Rubisco activity, and/or mesophyll conductance) are characterized by the Diurnal Water:Carbon Index (DWCI), which measures the degree of coupling between ET and gross primary productivity (GPP) within each day. As a proof of concept, the metrics indicated morning shifts and decoupling effects at 6 European sites during the 2003 heatwave event. Globally, we found indications of hydraulic limitations in the form of significantly high frequencies of morning shifted days in dry/Mediterranean climates and savanna/evergreen plant functional types (PFT), whereas high frequencies of decoupling were dominated by dry climates and grassland/savanna PFTs indicating a prevalence of non-stomatal limitations in these ecosystems. Overall, both the diurnal centroid and DWCI were associated with high net radiation and low latent energy typical of drought. Using three water use efficiency (WUE) models, we found the mean differences between expected and observed WUE to be -0.09 to 0.44 umol/mmol and -0.29 to -0.40 umol/mmol for decoupled and morning shifted days respectively compared to mean differences -1.41 to -1.42 umol/mmol in dry conditions. These results suggest that morning shifts/hydraulic responses are associated with an increase in WUE whereas decoupling/non-stomatal limitations are not.

4. The Introduction section is organized by sub-sections. The three levels of organization is pretty unusual for scientific papers. I suggest that the authors remove sub-section titles and rewrite it as a regular Introduction. If the authors intend to provide more background material, it is better to add a Background section right after the Introduction.

   Based on the reviewers suggestion, the sub-sections have been removed and the Introductions has been further shortened to be more focused on the subject at hand, such as removing the focus on transpiraiton estimates as suggested by reviewer 1 (comment 3), as well as removing discussion of the iWUE metrics which hopefully alleviates some unnecessary confusion.

5. It is unclear which FLUXNET database (LaThuile or 2015) is used in this study. Details should be provided. The Baldocchi et al. 2008 paper does not seem to be the proper citation for the database used. The supplementary figure can be moved to the manuscript given the relatively small number of illustrations.

   We agree that this was not made entirely clear and the reference was not the most appropriate one. We have updated our references to include the citation of the dataset pointing to the actual download location.

Further, as we have expanded the results section based on the reviewers suggestions, we have increased the number of figures from 5 to 7, with an additional supplementary figure showing a sensitivity analysis of the metrics to GPP and air temperature, as well as sensitivity of the frequency decoupled and morning shifted days based on different thresholds.

6. Line 30, page 4: FileS1 is a nice way to present study sites. Adding some technical details about how this kind of file was created will be helpful to the audience who might want to produce this kind of illustration as well.

   We are glad the reviewer likes our presentation of the sites. The file is created using the Bokeh package in Python based on the following technical example: https://bokeh.pydata.org/en/latest/docs/user_guide/geo.html. A reference to this example has been added to File S1, hopefully allowing these types of presentations to be more commonplace in literature.

7. Figure 2: An overall title should be added for the figure. Moreover, "upper" and "lower" should be changed to something like "Upper panels:" and "Lower panels:", respectively.

   We have given this figure the overall title of "Theoretical overview of diurnal water carbon index" and the upper and lower designations have been changed as the reviewer suggested. This designation was also used in the new figures (now figures 3 (R1) and 5 (R2))

8. Line 8, page 3: "are GPP" should be changed to "GPP are".

   This has been corrected.

9. Line 25, page 8: remove "be".

   This has been corrected.

**References**

[1] Ph. Ciais et al. "Europe-wide reduction in primary productivity caused by the heat and drought in 2003". en. In: *Nature* 437.7058 (Sept. 2005), pp. 529–533. ISSN: 0028-0836, 1476-4687. DOI: 10.1038/nature03972. URL: http://www.nature.com/articles/nature03972 (visited on 01/16/2018).

[2] A. Granier et al. "Evidence for soil water control on carbon and water dynamics in European forests during the extremely dry year: 2003". en. In: *Agricultural and Forest Meteorology* 143.1-2 (Mar. 2007), pp. 123–145. ISSN: 01681923. DOI: 10.1016/j.agrformet.2006.12.004. URL: http://linkinghub.elsevier.com/retrieve/pii/S0168192306003911 (visited on 01/16/2018).

[3]  Ashley M. Matheny et al. "Characterizing the diurnal patterns of errors in the prediction of evapotranspiration by several land-surface models: An NACP analysis: Error patterns in modeled transpiration". en. In: *Journal of Geophysical Research: Biogeosciences* 119.7 (July 2014), pp. 1458–1473. ISSN: 21698953. DOI: 10.1002/2014JG002623. URL: http://doi.wiley.com/10.1002/2014JG002623 (visited on 02/15/2016).

[4]  M. Reichstein et al. "Reduction of ecosystem productivity and respiration during the European summer 2003 climate anomaly: a joint flux tower, remote sensing and modelling analysis". en. In: *Global Change Biology* 13.3 (Mar. 2007), pp. 634–651. ISSN: 1354-1013, 1365-2486. DOI: 10.1111/j.1365-2486.2006.01224.x. URL: http://doi.wiley.com/10.1111/j.1365-2486.2006.01224.x (visited on 01/16/2018).

[5]  Kell B. Wilson et al. "Diurnal centroid of ecosystem energy and carbon fluxes at FLUXNET sites: DIURNAL ENERGY FLUXES AT FLUXNET SITES". en. In: *Journal of Geophysical Research: Atmospheres* 108.D21 (Nov. 2003). ISSN: 01480227. DOI: 10.1029/2001JD001349. URL: http://doi.wiley.com/10.1029/2001JD001349 (visited on 06/13/2016).

---

## Author Comment (AC3) · 23 Jan 2018

**Response to enteractive comment by Anonymous Referee #3 on "Water stress induced breakdown of carbon-water relations: indicators from diurnal FLUXNET patterns"**

Overall, I am supportive of the goals of this study. I agree that the asymmetry of the diurnal cycle of ET, and the correlation between GPP and ET, likely contain meaningful information about ecosystem response to drought. I also applaud the author's efforts to link these metrics to insights informed by mechanistic theory.

1. The analysis presented here is very broad (i.e. results are synthesized across many sites, and often many PFTs). Cross-site syntheses like this are invaluable for understanding broad patterns in vegetation functioning; however, the objectives of this study would benefit from a more in-depth analysis of the results at a least few sites, preferably sites where there exist independent estimates of plant water relations during periods of hydrologic stress (i.e. from gas exchange, sap flux, isotope analysis of tree cores, etc). This sort of analysis would give the readers confidence that the C_ET and DWCI metrics are really reflecting stomatal and non-stomatal limitations to transpiration and carbon uptake, and aren't unduly contaminated by the many sources of uncertainty in using eddy covariance measurements to infer GPP and water use efficiency (e.g. Knauer et al. 2017). Or, to put it differently the new metrics introduced in this manuscript merit some "proof of concept" before they are applied broadly.

> As the reviewer has pointed out, the presented analysis is indeed takes a broad approach to ecosystem physiology. This broad approach was undertaken in part due to the lack of a congruent dataset of independent estimates of plant water relations with which to compare the FLUXNET dataset. While sites do exists with measurements such as sap flux and isotope composition, they are currently not well homogenized and synchronizes in a way that is conducive to do an analysis across ecosystems, though some are definitely in the pipeline such as the SAPFLUXNET initiative. However, in an effort to give the presented manuscript a more in-depth analysis, we have included a new section in the manuscript which focuses on 6 European sites and compares average monthly values to the monthly values in 2003 when the continent was hit by a heat wave and drought. As seen in Figure 3 (here R1), the sites show diverse responses in the summer of 2003, with some exhibiting distinct morning shifts and some decoupling, but each of the sites show some response to the high temperatures and low water conditions of the heatwave. This new figure is accompanied by the following text in the Results section:

>> As a case study, $C_{ET}^*$ and DWCI time-courses for eight sites from Europe are shown in Figure 3, with an emphasis on 2003 when the continent was struck by a heatwave that was shown to effect both the carbon and water cycles [1, 9, 2]. For DWCI, forest sties showed high water:carbon coupling throughout the growing season, with the exception of Peuchebon (FR-Pue) which showed

[Figure]

Figure R 1: Monthly median diurnal water carbon index (DWCI, lower panels) and diurnal centroids ($C^*_{ET}$, upper panels) for 6 sites in Europe. Data from all years available (black) is compared to 2003 (red) during which a drought event resulted in high temperatures and low precipitation throughout the summer. Note DWCI of 0-100 indicate lowest-highest probability of diurnal carbon:water coupling and $C^*_{ET}$ of -1-1 indicate one hour morning shifted-one hour afternoon shifted ET. Vertical bars represent interquartile range. Sites from 5 plant functional types: evergreen broadleaf (EBF), deciduous broadleaf (DBF) and evergeen needleleaf (ENF) forests, as well as grasslands (GRA). Ecosystems show tendancies of morning shifts (e.g. DK-Sor and IT-Mal) and carbon:water decoupling (e.g. ES-ES1 and HU-Bug) during the drought year.

a regular seasonal cycle of decoupling. The grassland site (HU-Bg) showed a higher variability in DWCI compared to the forest sites (all others). All sites showed either a decrease in median DWCI or an increase in variability during 2003, generally in July or August, particularly at Hainich (DE-Hai), Bugacpuszta (HU-Bug), and El Saler (ES-ES1). This increase in decoupling during 2003 is consistent with the hypothesis of non-stomatal limitations being expressed in hot, dry conditions. Median diurnal centroid values across all years varied in absolute magnitude, but were generally near or above zero, i.e. the water cycle showed no shift or an afternoon shift. One exception would be the Mediterranean oak forest of Puechabon, which shows a slight seasonal cycle of morning shifts going from a slight afternoon shift to a slight morning shift during June, July, and August. During drought years, sites that showed distinctive morning shifts were Puechabon (FR-Pue), Soroe (DK-Sor), and Loobos (NL-Loo). The framework that morning shifts are associated with water stress from soil moisture depletion would be supported by the increase in morning shifts during 2003, though factors such as species composition and access to soil water would play a significant factor and could account for the differences among sites. All sites which had significantly different (p<0.05, Wilcoxon rank-sum test) DWCI values between 2003 and all other years except Puechabon, whereas with $C^*_{ET}$ only Puechabon, Soroe, and Loobos showed significant differences.

Furthermore, we hope that by demonstrating the response of the metrics across a large number of sites, and providing the equations and associated code, we can provide the tools to those with both the access

and expertise of these particular sites with independent measurements. In this regard, we have taken the suggestion given by reviewer #1 to give a direct call for such an analysis, with the following text being added to the Discussion section "Looking beyond sums and means":

> Given the broad nature of the analysis here, the metrics and hypothesis presented would benefit from site specific validations such as looking to see if the morning shits and decoupling are indeed associated with lower soil moisture levels, leaf water potentials, and/or decreases in sap flux. Sap flux in particular could give some interesting insights, as the diurnal patters in sap flux velocity will also have an offset to incoming radiation related to tree capacitance, therefore relating sap flow diurnal centroids to the ET diurnal centroid could give some information on changes in plant water recharge. Furthermore, the diurnal centroid base metrics complement the hysteresis quantification methods such as those employed by Zhou et al. [13] and Matheny et al. [7], with the advantage of $C_{ET}^*$ being compensation for cloudy conditions and possibly less influence of noise, though an intercomparison would be useful to explore the strengths and weaknesses of the different approaches. By providing both the equations and related code of the metrics, we the authors hope the metrics will be used by the community for both validation and to further ecophysiological understanding.

> Finally, we have added Figure 5 (R2) highlighting the tree vs. grass responses based on evaporative fraction, which shows both the effect of decoupling and morning shifts in forest, savanna, and grassland sites with lower evaporative fraction, as well as showing the divergent responses. We hope these added analyses will give the reader confidence that the metrics are reflecting water stress responses.

2. I also had a few concerns about the presentation and interpretation of the water use efficiency theory. First, the authors attributed the afternoon decline in ET to "hydraulic limitation" driven specifically by challenges of moving liquid water from roots to the leaves as soil dries (e.g. Lines 24-25). While I agree that hydraulics are an important control on stomatal functioning, stomates may also close directly in response to rising VPD even if soil moisture is unchanged (as discussed at length in the stomatal optimization literature), and the mechanisms responsible for the VPD response are still not yet clear. Thus, it may be more appropriate to describe the afternoon decline in ET as simply "stomatal limitations."

> The reviewer makes a very good point and highlights the complexities of the problem. Indeed, the key control mechanism a plant has in the soil-plant-atmosphere continuum is via the stomates, especially at the diurnal scale. Likewise, we agree that the stomates have been shown to have a primary response to VPD, indeed we introduce the $VPD^-0.5$ term when calculating the DWCI in an effort to mitigate the VPD effects on decoupling diurnal GPP and ET. However, the closure of stomates due to VPD would cause an afternoon decrease in GPP but not necessarily in ET as VPD is also a driver of ET. In other words, as the stomate is closing in response to VPD, VPD is also pulling harder at the water in the leaf. So the

[Figure]

Figure R 2: Median diurnal water carbon index (DWCI, upper panel) and diurnal centroid ($C_{ET}^*$, lower panel) of plant flunctional types binned by evaporative Fraction (EF, low values indicate dry conditions). Note DWCI of 0-100 indicate lowest-highest probability of diurnal carbon:water coupling and $C_{ET}^*$ of -1-1 indicate one hour morning shifted-one hour afternoon shifted ET. Evergreen needleleaf (ENF), deciduous broadleaf (DBF), and evergreen boradleaf (EBF) forests show increased morning shifts (low $C_{ET}^*$) with decreasing EF when compared to grassland (GRA) sites which tended to have decreased carbon:water decoupling (low DWCI) with decreasing EF. Savanna ecosystems (SAV) show a high degree of decoupling and intermediate levels of morning shifts. Vertical bars represent interquartile range.

hypothesis driving the diurnal centroid is then that the plant is responding to some form of hydraulic limitation of moving water, with the assumption that if the plant had unlimited access to water, and the ability to move it to the leaves, it would continue to keep the stomates open and take up carbon (the carbon cost of water would be zero as water is infinite). Though the reviewer makes a good point and this is simply a hypothesis for the metrics. As a further test, we have added additional subplots to Figure 4 (Figure 6a,d in the new manuscript, here R3a,d) which shows the response (as color) along the axes of evaporative fraction (EF=ET/ETpot) and VPD.

[Figure]

Figure R 3: Mean DWCI (upper panels) and $C_{ET}^*$ (lower panels) with respect to evaporative fraction (EF) by vapor pressure deficit VPD (a,d), latent energy (LE) by Rn (b,e) and LE by GPP (c,g). Note DWCI of 0-100 indicate lowest-highest probability of diurnal carbon:water coupling and $C_{ET}^*$ of -1-1 indicate one hour morning shifted-one hour afternoon shifted ET. Points with high Rn and low LE are associated with both low DWCI and $C_{ET}^*$, indicating that both metrics are related to water limitations. Though both metrics are associated with low EF, DWCI shows a much higher response to atmospheric demand as measured by VPD, with $C_{ET}^*$ showing very limited response. Both metrics, and DWCI in particular, show low values with high ET and low Rn, though these points are also associated with over closed energy balances (LE+H>Rn-G). Both metrics are associated with low GPP, but the $C_{ET}^*$ is restricted to both low GPP and ET, indicating water and carbon can decouple over a wider range of water stress. This also holds when points with energy balance over-closer are excluded (data not shown).

From the new subplot (Figure 6a,d (R3a,d)), one can see that the diurnal centroid in fact is not very responsive to mean daily VPD, instead being almost entirely responsive to EF. This would support our hypothesis that the morning shift is significantly driven by hydraulic limitations rather then simply the VPD response. In contrast, the same subplots shows that the DWCI is much more responsive to high VPD, indicating that decoupling is happening during high atmospheric demand. This is highlighted in the Results section with the text:

Apart from the response to periods of high LE and low Rn, the metrics showed diverging response when looking at EF (ET/PET which is similar to LE/Rn) and VPD, with DWCI showing a much stronger response to VPD and $C_{ET}^*$ showing a much stronger response to EF (Figure

6a,d). This difference in response would indicate that DWCI is more responsive to atmospheric demand (estimated via VPD) and $C^*_{ET}$ is more responsive to water limitations.

So while the reviewer is correct in that this morning shift likely would ultimately be a stomatal control, we hope to highlight this underlying hypothesis which differentiates what we are looking for from the stomatal limitations often describing VPD effects on carbon uptake. To clarify this point in the introduction, we have added the following text:

> Under this hydrauilc limitation framework a plant will be reacting to the inability to transport water, even though the key control mechanism for a plant is via the stomata, possibly expressed as an increase in sensitivity. Such assumptions are consistent with the mechanisms encoded in some land surface and ecosystem models, which account for water limitations by scaling the water to carbon ratio in relation to available soil moisture.

3. Next, the function ET=i*GPP*sqrt (VPD), proposed by Zhou et al. (2015), is referred to in this manuscript as the "Katul" model; presumably this nomenclature originates from the theory presented in Katul et al. (2010), which presents arguments leading to the equation: ET = GPP x sqrt(VPD) / sqrt(1.6*lambda*ca) The parameter lambda is the so-called "marginal water use efficiency" and ca is atmospheric CO2 concentration. This is similar to the Eq. 8 in the present manuscript: ET = i x GPP x sqrt(VPD) If i = 1/ sqrt(1.6*lambda*ca) I appreciate that the authors have attributed the model to Katul et al. (2010), who presented the theory on which the equation is based. However, before the authors attribute the model's "inability to make accurate predictions" to "be a result of a failure of their underlying assumptions," (Page 9, lines 10-11), care should be taken to make sure the underlying assumptions are properly stated and considered. The Katul et al. (2010) result relies on an assumption of Rubisco-limited photosynthesis (which generates a linear A-Ci curve). . .this assumption is not likely to hold in dense forests where understory vegetation is often light-limited. Second, and perhaps more importantly, stomatal optimization theory assumes that the parameter lambda should hold constant over timescales of hours, but varies over longer timescales (days to weeks) as other slowly-evolving boundary conditions change (Manzoni et al. 2013, Palmroth et al. 2013). So in that regard, I disagree with the author's assessment on Page 8, Line 5, that the "Katul" model "makes the assumption that the WUE is constant if corrected by the effect of VPD." The potential for lambda to vary may also help to explain the tendency of the Katul (and other) models to underestimate WUE during dry conditions (i.e. Figure 5), especially if the 'i' parameter is determined using observations from well-watered conditions.

> We would like to thank the reviewer for so succinctly summarizing what is not always made clear in many manuscripts, including this one in present state. In regard to the initial comments, the reference to Katul 2010 is one that should be cited in this context and this reference has now been included in the appropriate places. Furthermore, the text has been changed so that the model is now initially introduced as:
>
> > These carbon:water links are fundamental to understanding how stomata are regulated and

underly key functioning in mechanistic plant and ecosystem models. One such set of models are those based on optimality theory which posit that plants tend to optimize carbon gains to water losses, such as those described by Katul et al. [4] and Katul, Palmroth, and Oren [3]. These concepts from Katul, which carry the assumptions of RuBISCO limitaion, were built upon by Zhou et al. [13] and Zhou et al. [11] to give the equation,

$$uWUE = \frac{GPP \cdot \sqrt{VPD}}{ET}$$

where the $\sqrt{VPD}$ accounts for the stomatal response to vapor pressure deficit (VPD). Accounting for the VPD response allows for a more stable metric of WUE that is temporally more stable and physiologically more meaningful, such as when comparing the diurnal cycles of carbon and water.

The point that the carbon cost of water is likely to change is further highlighted in the Materials and Methods where the subsection "Models and parameter estimation" has been expanded to say,

> The "Katul" model, as defined and used in calculation of the DWCI, is based in stomatal optimization theory [4, 3, 11], which makes the assumption that the WUE is constant if corrected by the effect of VPD, using an inverse square root as the assumed relationship. Though the constant nature of uWUE may not be correct, with the optimal carbon cost of water changing over day or weeks [5, 8], a yearly parameter of uWUE was estimated which is consistent with other modeling exercises [12].

Note that an explanation of how these yearly parameters are calculated follows this statement, which hasn't changed from the original submission. Finally, as the reviewer has pointed out, our statement that biases of the Katul model are a failure of underlying assumptions becomes misleading as we do not reference what assumptions we have made. As such, this has been amended to say:

> Both the Katul and Boese models are theoretically based and here implemented have the underlying assumptions of RuBiSCO-limited conditions and constant carbon cost of water throughout the season which may not reflect reality.

4. Finally, when using the shape of the diurnal pattern of ET to infer stomatal limitation, I wondered why the authors focused only on the shift in the peak, and not the overall degree of asymmetry between morning and afternoon periods (for example, if the ET data from hours 0-12 are reflected about the solar noon axis, that is the area between the reflected and actual ET data).

This idea was actually explored during the course of the analysis while looking for a metric to quantify

the asymmetry of the diurnal patterns, and would lead in the direction of hysteresis analysis that other researchers had done such as Zhou et al. [13] and Matheny et al. [7]. One issue with this type of analysis is that one tends to compare individual half hours of the morning to half hours in the evening which can be problematic. For example, in the case of the example the reviewer suggests, a cloud passing in afternoon would be indistinguishable to a physiological response to water limitation as both would reduce the water flux. So by using the diurnal centroid of ET vs Rg we are able to measure the asymmetry while correcting for changes in incoming energy. Previous experiences with hysteresis approaches showed that the quantifications could be very sensitive to noise, especially at the daily scale and the diurnal centroid may be a more robust metric. However, the reviewer makes a good point that hysteresis quantification and diurnal centroid based metrics are two approaches to explore the same effect, a point which we have added to the discussion:

> Furthermore, the diurnal centroid base metrics complement the hysteresis quantification methods such as those employed by Zhou et al. [13] and Matheny et al. [7], with the advantage of $C_{ET}^*$ being compensation for cloudy conditions and possibly less influence of noise, though an intercomparison would be useful to explore the strengths and weaknesses of the different approaches.

A few other comments include:

5. Page 2, lines 15-24. The presentation of the iWUE models would benefit from a more general explanation of what each of the three metrics actually describes and/or is sensitive too. E.g. WUE is useful for understanding broad patterns of ecosystem water use and carbon uptake, but is sensitive to non-biological drivers (e.g. VPD). The iWUE attempts to correct for the direct effect of VPD on transpiration, and is thus a more biologically relevant metric. The uWUE further attempts to correct for stomatal closure under high VPD, and therefore may be more closely linked to the "non-stomatal" limitations to gas exchange during drought.

> In an effort to make this section more clean and concise, we have removed the reference to Beer et al. 2008 and iWUE as it was not further referenced in the manuscript. We have also taken the reviewers advice and expanded the section to give some context to the VPD term. The text now reads:

>> These carbon:water links are fundamental to understanding how stomata are regulated and underly key functioning in mechanistic plant and ecosystem models. One such set of models are those based on optimality theory which posit that plants tend to optimize carbon gains to water losses, such as those described by Katul et al. [4] and Katul, Palmroth, and Oren [3]. These concepts from Katul, which carry the assumptions of RuBISCO limitation, were built upon by Zhou et al. [13] and Zhou et al. [11] to give the equation,

$$uWUE = \frac{GPP \cdot \sqrt{VPD}}{ET}$$

where the $\sqrt{VPD}$ accounts for the stomatal response to vapor pressure deficit (VPD). Accounting for the VPD response allows for a more stable metric of WUE that is temporally more stable and physiologically more meaningful, such as when comparing the diurnal cycles of carbon and water.

6. Page 4, Line 17: "the use of EC measured diurnal patterns of carbon, water, and energy fluxes to derive clues on ecosystem drought responses at a daily resolution could prove valuable, if nothing less than a benchmark to test current hypotheses." As far as rationale for the work goes, I found this to be rather weak. Perhaps the authors could give specific examples of hypotheses that could be tested with these metrics.

> We agree that this statement was rather vague, it has now been changed to be more explicit:
>
> > In this sense, the use of EC measured diurnal patterns of carbon, water, and energy fluxes to derive clues on ecosystem drought responses at a daily resolution could prove valuable both as a means to identify potential periods of ecosystem stress, inform machine learning algorithms on ecophysiological conditions not found in environmental variables, as well as benchmarking a models ability to capture sub-daily dynamics.

7. Page 9, line 20: "sites under water stress tended to have C_ET < -0.50." How do the authors know that the sites were under water stress? This gets back to my original point about validating the metrics against independent observations of plant function.

> The reviewer makes a good point, one which was also brought up by reviewer 1 (comment 9). As discussed in comment 1 of this review, evidence of metric response is seen in the case study of the European heatwave of 2003. Furthermore, to address the appropriateness of these thresholds, we have included a new sensitivity analysis which shows the relationship of frequency of uncoupled and morning shifted days based on different threshold levels (Figure S2. here R4). This analysis indicates that indeed these levels may have been slightly too strict and have been changed to DWCI<25 and $C_{ET}^*$<-0.25, which gives a stronger response while does not change the patterns. The unchanged patters demonstrate both the robustness of the patterns, as well as that the absolute threshold levels are flexible with acceptable ranges of about 5-75 for DWCI and -1.0-0.0 for $C_{ET}^*$. This analysis is now discussed in the results section with the following text:
>
> > These thresholds were chosen to highlight frequency differences between sites and were shown to have large metric responses under dry conditions while having low frequencies under wetter conditions (see sensitivity analysis in supplementary figure S2). Furthermore, these thresholds

results in a similar median frequency of uncoupled and morning shifted days between all site-years being 8.7% and 9.4% of days respectively. The similarity in median frequencies across site-years allowed for easier inter-comparison between the two metrics. The frequency of de-coupling and morning shifts using these thresholds for each site can be found in the map found in File S1.

8. Figure 3: The text is small and hard to read.

   The text size has been increase to improve readability (Figure R5).

9. Page 13, Lines 15-25: I found this discussion of the links between C_ET and isohydricity to be highly speculative, notably because isohydricity tends to describe plant response to declining soil moisture, yet the afternoon stomatal closure may be largely caused by increasing VPD.

   To further the points discussed above, the VPD responses are likely tied to the concept of isohydricity. The review by Martínez-Vilalta and Garcia-Forner [6] makes the point that VPD and transpiration dynamics feed back into the rate of soil moisture depletion and how fast a plant will reach the point of hydraulic failure. Discussion of these points have been added to highlight these feedbacks, and the section now reads:

[revised manuscript text omitted]